# UNIFIED BREAKDOWN ANALYSIS FOR BYZANTINE ROBUST GOSSIP

## ABSTRACT

Distributed approaches have many computational benefits, but they are vulnerable to attacks from a subset of devices transmitting incorrect information. This paper investigates Byzantine-resilient algorithms in a decentralized setting, where devices communicate directly with one another. We investigate the notion of *breakdown point*, and show an upper bound on the number of adversaries that decentralized algorithms can tolerate. We introduce an algorithmic framework that recovers ClippedGossip and NNA, two popular approaches for robust decentralized learning, as special cases. This framework allows us to generalize NNA to sparse graph, and introduce $CG^+$, which is at the intersection of the two. Our unified analysis framework gives near-optimal guarantees for $CG^+$ (and other approaches with additional assumptions). Experimental evidence validates the effectiveness of $CG^+$ and the gap with NNA, in particular against a novel attack tailored to sparse graphs that we introduce.

## 1 INTRODUCTION

Distributed machine learning, in which the training process is performed on multiple computing units (or nodes), responds to the increasingly distributed nature of data, its sensitivity, and the rising computational cost of optimizing models. While most distributed architectures rely on coordination from a central server, some communication networks favor peer-to-peer exchanges, and global coordination can be costly in these cases. Besides, the decentralized setting has several other perks: it mitigates the communication bottleneck and failure risk at the main server, and provides additional privacy guarantees since agents only have a local view of the system (Cyffers et al., 2022). However, distributing optimization over a large number of devices introduces new security issues: software may be faulty, local data may be corrupted, and nodes can be hacked or even controlled by a hostile party. Such issues are modeled as *Byzantine* node failures (Lamport et al., 1982), defined as omniscient adversaries able to collude with each other.

Standard distributed learning methods are known to be vulnerable to Byzantine attacks (Blanchard et al., 2017), which has led to significant efforts in the development of robust distributed learning algorithms. From the first works tackling Byzantine-robust SGD (Blanchard et al., 2017; Yin et al., 2018; Alistarh et al., 2018; El-Mhamdi et al., 2020), methods have been developed to tackle stochastic noise using Polyak momentum (Karimireddy et al., 2021; Farhadkhani et al., 2022) and mixing strategies to handle heterogeneous loss functions (Karimireddy et al., 2020; Allouah et al., 2023). In parallel to these robust algorithms, efficient attacks have been developed to challenge Byzantine-robust algorithms (Baruch et al., 2019; Xie et al., 2020). To bridge the gap between algorithm performance and achievable accuracy in the Byzantine setting, tight lower bounds have been constructed for the heterogeneous setting (Karimireddy et al., 2020; Allouah et al., 2024). Yet, all these works rely on a trusted central server to coordinate training.

In contrast, the decentralized case has been less explored. In particular, it is still unclear how many Byzantine nodes can be tolerated over a given communication network before aggregation protocols fail. In fact, the network is often assumed to be fully connected (El-Mhamdi et al., 2021; Farhadkhani et al., 2023), and most papers that go beyond this assumption by addressing sparse graphs either do not give clear convergence rates or give weak guarantees on the asymptotic error (Peng et al., 2021; Fang et al., 2022; Wu et al., 2023). While criteria for using SGD with generic robust decentralized aggregation rules have been proposed (Wu et al., 2023; Farhadkhani et al., 2023), Decentralized

| Status | Algorithm | Setup | Breakdown Point | Experiments |
|---|---|---|---|---|
| He et al. (2022) | ClippedGossip w. *oracle rule* | Gossip - not implementable | $b \leq \mathcal{O}(\gamma \mu_{\min})$ | none |
| He et al. (2022) | ClippedGossip w. adaptive rule | Gossip | No guarantee | Competitive |
| Farhadkhani et al. (2023) | NNA | Centralized case only | No guarantee for sparse graphs | None on sparse graphs |
| **New (theory)** | Clipped Gossip w. *oracle rule* | Gossip - not implementable | $b \leq \frac{\mu_{\min}}{8}$ | none |
| **New (algo + theory)** | Gossip NNA | Gossip - practical rule | $b \leq \frac{\mu_{\min}}{8}$ | Competitive yet worse breakdown |
| **New (algo + theory)** | CG+ | Gossip + practical clipping rule | $b \leq \frac{\mu_{\min}}{4}$ | Competitive |
| **New (algo + theory)** | CG+, *oracle rule* | Gossip - not implementable | $b \leq \frac{\mu_{\min}}{2}$ **(optimal)** | none |

Table 1: Summary of our results and comparison to previous work.

SGD suffers from the same flaws. For instance, while $(\alpha, \lambda)$-reduction (Farhadkhani et al., 2023) is agnostic to the communication network, NNA, the associated robust communication scheme, is only introduced and analyzed for fully-connected networks. ClippedGossip (He et al., 2022), which consists in clipping the standard *gossip averaging* updates (Boyd et al., 2006) used for decentralized mean estimation, is designed for sparse networks. Unfortunately, its clipping threshold requires inaccessible information about how many honest nodes are clipped, and the theory only applies for a small fraction of Byzantine failures.

Our work revisits both the ClippedGossip and NNA frameworks to solve the aforementioned shortcomings. To do so, we carefully study the decentralized mean estimation problem. This seemingly simple problem retains most of the difficulty of handling Byzantine nodes while allowing us to derive strong convergence and robustness guarantees. We then tackle general (smooth non-convex) optimization problems through a reduction. Our contributions are summarized in Table 1, and the key points are the following:

**1 - New upper bound on the breakdown point of robust algorithms.** We show that in general, Byzantine robust algorithms fail arbitrarily if the number of Byzantine neighbors per node exceeds a given threshold. This threshold is expressed in terms of spectral quantities of the graph (the *algebraic connectivity*), and recovers usual ones for fully-connected topologies.

**2 - Unified algorithmic framework.** We propose a unified algorithmic and analysis framework that provides near-optimal breakdown guarantees for ClippedGossip and NNA and allows to naturally introduce Gossip-NNA as well as $CG^+$, a more robust algorithm.

**3 - Stronger decentralized attacks.** We propose a theoretically grounded attack, called *Spectral Heterogeneity*, specifically designed to challenge decentralized algorithms by leveraging spectral properties of the communication graph.

## 2 BACKGROUND

### 2.1 DECENTRALIZED OPTIMIZATION.

We consider a system composed of $m$ computing units that communicate synchronously through a communication network, which is represented as an undirected graph $\mathcal{G}$. We denote by $\mathcal{H}$ the set of honest nodes, and $\mathcal{B}$ the (unknown) set of Byzantine nodes. Each unit $i$ holds a local parameter $\boldsymbol{x}_i \in \mathbb{R}^d$, a local loss function $f_i : \mathbb{R}^d \to \mathbb{R}$, and can communicate with its neighbors in the graph $\mathcal{G}$. We denote the set of neighbors of node $i$ by $n(i)$ and by $n_{\mathcal{H}}(i)$ (resp. $n_{\mathcal{B}}(i)$) the set of honest (resp. Byzantine) ones. We study decentralized algorithms for solving

$$\arg\min_{\boldsymbol{x} \in \mathbb{R}^d} \left\{ f_{\mathcal{H}}(\boldsymbol{x}) := \frac{1}{|\mathcal{H}|} \sum_{i \in \mathcal{H}} f_i(\boldsymbol{x}) \right\}. \tag{1}$$

Due to the averaging nature of Equation (1), centralized algorithms for solving this problem rely on global averaging of the gradients computed at each node. In the decentralized setting, we rely on (local) inexact averaging instead.

**Gossip Communication.** Standard decentralized optimization algorithms typically rely on the so-called *gossip* communication protocol (Boyd et al., 2006; Nedic & Ozdaglar, 2009; Scaman et al., 2017; Kovalev et al., 2020). The gossip protocol consists in updating parameters of any node $i$ with a linear combination of the parameters of its neighbors, with updates of the form $\boldsymbol{x}_i^{t+1} = \boldsymbol{x}_i^t - \eta \sum_{j=1}^m \boldsymbol{W}_{ij} \boldsymbol{x}_j^t$, where $\eta \geq 0$ denotes a communication step-size. The matrix of the weights $\boldsymbol{W} = (\boldsymbol{W}_{ij})_{i,j}$ is called the *gossip matrix*, and naturally defines the communication graph $\mathcal{G}$, in the sense that $\boldsymbol{W}_{ij} = 0$ if nodes $i$ and $j$ are not neighbors. By considering the matrix of honest parameters $\boldsymbol{X} = (\boldsymbol{x}_1, \ldots, \boldsymbol{x}_m)^T$, the gossip update is also conveniently written as

$$\boldsymbol{X}^{t+1} = \boldsymbol{X}^t - \eta \boldsymbol{W} \boldsymbol{X}^t. \tag{2}$$

In this work, we instantiate gossip algorithms by using as gossip matrix the weighted Laplacian matrix of the graph. More specifically, we consider a series of weights $(w_{ij})_{j>i}$ associated with each edge of the graph, and define the Laplacian matrix as $\boldsymbol{W}_{ii} = -w_{ij}$ for $i \neq j$, and $\boldsymbol{W}_{ij} = \sum_{j\neq i} w_{ij}$. The Laplacian matrix is symmetric non-negative, and its rows and columns sum to 0. To ease reading, we consider that the weights $w$ are part of the graph $\mathcal{G}$, so that a unique Laplacian matrix is associated with each graph $\mathcal{G}$. Thus, we denote by $\mu_{\max}(\mathcal{G}_{\mathcal{H}})$ and $\mu_2(\mathcal{G}_{\mathcal{H}})$ the largest and smallest non-zero eigenvalues of the Laplacian matrix $\boldsymbol{W}_{\mathcal{H}}$ of the *honest subgraph* $\mathcal{G}_{\mathcal{H}}$, and by $\gamma = \mu_2(\mathcal{G}_{\mathcal{H}})/\mu_{\max}(\mathcal{G}_{\mathcal{H}})$ its *spectral gap*. Spectral properties of the gossip matrix are known to characterize the convergence of gossip optimization methods. For instance, in the absence of Byzantine nodes, the plain gossip update with step-size $\eta \leq \mu_{\max}(\mathcal{G})^{-1}$ leads to a linear convergence of the nodes parameter values to the average of the initial parameters: $\|\boldsymbol{X}^t - \overline{\boldsymbol{X}}^0\|^2 \leq (1 - \eta\mu_2(\mathcal{G}))^t \|\boldsymbol{X}^0 - \overline{\boldsymbol{X}}^0\|^2$, for $\overline{\boldsymbol{X}}^0$ the matrix with columns $m^{-1}\sum_{j=1}^m \boldsymbol{x}_j^0$.

**Robustness Issue.** Gossip communication relies on updating nodes parameters by performing non-robust local averaging. As such, similarly to the centralized case, any Byzantine neighbor of node $i$ can drive the update to any desired value (Blanchard et al., 2017). Then, the poisoned information spreads through gossip communications.

## 2.2 Byzantine robust optimization.

**Threat model.** We consider Byzantine nodes to be omniscient adversaries, able to collude and to send distinct values to each of their neighbors. We measure their influence by considering the weight of Byzantine nodes in the neighborhood of each honest nodes, $\{w(n_{\mathcal{B}}(i)) = \sum_{j\in n_{\mathcal{B}}(i)} w_{ij}; i \in \mathcal{H}\}$ (as in LeBlanc et al. (2013); He et al. (2022)), instead of the total number of Byzantine nodes $|\mathcal{B}|$ as it is done for the centralized or fully-connected setting.

Similarly, the total number of honest nodes does not provide relevant information anymore, as the results depend on *how* they are linked, i.e., the topology of the honest subgraph. Therefore, for sparse topologies, we need to make an assumption about some property of the graph related to its topology, instead of simply the number honest neighbors. In the remainder of this paper, we consider spectral properties of the Laplacian of the honest subgraph as a relevant quantity for robustness analyses. Yet, we emphasize that our results on the breakdown depend on the spectral properties *of the honest subgraph*, meaning that for a given graph, these properties change depending on the location of Byzantine nodes. We introduce the following class of graphs to take this dependence into account.

**Definition 1.** *For any $\mu_{\min} \geq 0$ and $b \in \mathbb{N}$, we define the class of graphs (with edge weights)*

$$\Gamma_{\mu_{\min},b} = \{\mathcal{G}, w \ s.t. \ \mu_2(\mathcal{G}_{\mathcal{H}}) \geq \mu_{\min} \ and \ \max_{i\in\mathcal{H}} w(n_{\mathcal{B}}(i)) \leq b\}.$$

In other words, we introduce a subset of all possible graphs, partitioning in terms of (i) their algebraic connectivity, that is restricted to be larger than a minimal value $\mu_{\min}$, and (ii) the maximal total Byzantine neighbors weight of an honest node, which is restricted to be smaller than $b$.

Note that if $w_{ij} = 1$ for all $i > j$, then $w(n_{\mathcal{B}}(i)) = |n_{\mathcal{B}}(i)|$, which is why we will sometimes abuse language and refer to the 'number of Byzantine neighbors' instead of the sum of their weights. One should read Definition 1 as a sparse graph extension of the standard *"there are at most $b$ byzantine nodes and at least $|\mathcal{H}|$ honest ones"*, which now involves the relative positions of Byzantine and honest nodes in the graph and the edge weights. For given $|\mathcal{B}|$ and $\mu_{\min} \geq 0$, depending on the location of the Byzantine nodes, a given graph topology can either fall in $\Gamma_{\mu_{\min},b}$ (if Byzantine nodes are "well-spread"), or not (if they are adversarially chosen).

**Approximate Average Consensus.** The average consensus problem consists in finding the average of $m$ vectors locally held by nodes. It is a specific case of Equation (1) obtained by considering $f_i(\boldsymbol{x}) = \|\boldsymbol{x} - \boldsymbol{y}_i\|^2$. Due to adversarial attacks, some *bias* is introduced by Byzantine nodes during aggregation steps, so only an *approximate* solution of the average of honest nodes vector $\overline{\boldsymbol{y}}_{\mathcal{H}} := |\mathcal{H}|^{-1} \sum_{i \in \mathcal{H}} \boldsymbol{y}_i$ can be expected. In the centralized and fully-connected settings, the variance between nodes can be reduced to 0 in one step (consensus is reached). In the (gossip) decentralized setting, consensus requires multiple communications. At each, we aim for *reducing the variance* at the cost of some bias. Guarantees given in Section 4 exactly reflect this trade-off: they quantify how much variance reduction is obtained at each step, and at the cost of what bias. Note that variance reduction here is to be understood as how different the parameters of the various nodes are, and is not directly linked with the variance of stochastic gradients. We now introduce the $\alpha$-robustness of a communication algorithm on a graph.

**Definition 2** ($\alpha$-robustness on $\mathcal{G}$.)**.** *For any $\alpha < 1$, a communication algorithm $A$ is $\alpha$-robust on a graph $G$ if from any initial local parameters $\{\boldsymbol{x}_i; i \in \mathcal{H}\}$, it allows any honest node $i$ to compute a vector $\hat{\boldsymbol{x}}_i$ such that*

$$\frac{1}{|\mathcal{H}|} \sum_{i \in \mathcal{H}} \|\hat{\boldsymbol{x}}_i - \overline{\boldsymbol{x}}_{\mathcal{H}}\|^2 \le \alpha \frac{1}{|\mathcal{H}|} \sum_{i \in \mathcal{H}} \|\boldsymbol{x}_i - \overline{\boldsymbol{x}}_{\mathcal{H}}\|^2.$$

Imposing $\alpha < 1$ means that we would like to be closer to the initial solution after the aggregation step than before (the variance reduction needs to be larger than the bias we introduce). Note that $\alpha = 1$ can trivially be achieved by not communicating at all. Remark that the $\alpha$-robustness of an algorithm on a graph $\mathcal{G}$ means that a *single step* of the algorithm strictly reduces the average quadratic error. However, it does not mean that multiple steps would result in a geometric decrease, indeed, we cannot simply use induction as $\hat{\overline{\boldsymbol{x}}}_{\mathcal{H}} \ne \overline{\boldsymbol{x}}_{\mathcal{H}}$. In the following, we show that $\alpha$-robustness on all graphs in $\Gamma_{\mu_{\min}, b}$ cannot be achieved for all values of $\mu_{\min}$ and $b$.

# 3 FUNDAMENTAL LIMITS OF DECENTRALIZED COMMUNICATION SCHEMES

In this section we provide an upper bound on the number of Byzantine neighbors that can be tolerated by any algorithm running on a communication network in which the honest subgraph has a given algebraic connectivity.

**Theorem 1.** *For any $\mu_{\min} \ge 0$, $b \ge 0$, if $\mu_{\min} \le 2b$, then for any $H \ge 2b$, there exists $\mathcal{G} \in \Gamma_{\mu_{\min}, b}$ with $H$ honest nodes such that no communication algorithm can be $\alpha$-robust on $\mathcal{G}$.*

*Sketch of proof.* The proof relies on considering a specific graph $G_{H,b}$, decomposed as three cliques of $H/2$ nodes, for an even $H$ such that $H/2 \ge b$. We choose that nodes in any of the three cliques is neighbor to exactly $b = k$ nodes in each of the two other cliques, in circular order. Finally, we assume that two of the three cliques are honest, and the third one is composed of Byzantine nodes.

First, $G_{H,b} \in \Gamma_{\mu_{\min}, b}$. Indeed, the Laplacian matrix of the honest subgraph has an algebraic connectivity of $2b$, i.e. $\mu_2((G_{H,b})_{\mathcal{H}}) = 2b$, and $2b \ge \mu_{\min}$ by our assumption. Moreover, each honest node has exactly $b$-Byzantine neighbors. Second, we show that no algorithm can be $\alpha$-robust on $G$. Indeed, for any of the two honest cliques, the Byzantine clique is indistinguishable from the other honest clique. As such, Byzantine nodes can send parameter values such that honest nodes cannot hope to improve the global error in general. We refer the reader to Appendix C.1 for details. $\qquad\square$

It follows from Theorem 1 that, when aiming at obtaining a theoretical guarantee that quantifies the robustness of the honest graph through $\mu_{\min}$, we must have $\mu_{\min} > 2b$. In the specific case of a fully connected graph with unitary weights, where $\mu_{\min} = |\mathcal{H}|$, this condition boils down to requiring $|\mathcal{H}| > 2|\mathcal{B}|$, which is aligned with common robustness criteria for distributed system (Lamport et al., 1982; Vaidya et al., 2012; El-Mhamdi et al., 2021). As we show in Section 4, this upper bound on $b$ is tight up to a factor 2, in the sense that there exists an $\alpha$-robust communication scheme over all $G \in \Gamma_{\mu_{\min}, b}$ as soon as $4b \le \mu_{\min}$.

**Algebraic connectivity as a robustness criterion.** Note that Theorem 1 does not imply that all aggregation methods fail as long as $2b \ge \mu_{\min}$, but rather than since there exists a graph for which it is the case, one cannot prove that an aggregation method works with $b > \mu_{\min}/2$ Byzantine nodes for all graphs. Yet, one can still prove breakdown points using other graph-related quantities (which

might lead to tolerating $b > \mu_{\min}/2$ Byzantine nodes for some graph architectures), or restricting the graph topologies considered. This gap is standard in the optimization literature, since convergence results often involve the spectral properties of the gossip matrix, whereas they do not naturally appear when proving lower bounds on the number of iterations required to reach a certain accuracy. For instance, the convergence rates of decentralized optimization algorithms depend on the (square root of the) *spectral gap* of the Laplacian matrix of the communication graph (Scaman et al., 2017; Kovalev et al., 2020), whereas iteration lower bounds are proven in terms of diameter. Yet, these decentralized algorithms are termed as optimal as their guarantees match the lower bound on the path graph.

**The Approximate Consensus Problem and dimension-dependent breakdown points.** The design of algorithms aiming at finding the average of parameters within a communication network is related to the *approximate consensus problem* (ACP) (Dolev et al., 1986). In the standard ACP problem, nodes need to converge to the same value while remaining within the *convex hull* of initial parameters. Yet, communication-optimal methods for this problem are memory and computationally expensive (Fekete, 1986). More recently, the work of LeBlanc et al. (2013) aims at designing communication schemes that only use local information with computationally efficient aggregation rules. They show that standard robustness criterion of *connectivity* (Sundaram & Hadjicostis, 2010) does not properly reflect the robustness of a network for such methods. To mitigate this issue, they introduce the notion of $r$-robust networks. Vaidya (2014) generalizes this result by proving that the ACP cannot be solved using an algorithm with *iterative communication* on a system of $m$ nodes with $b$ Byzantine failures in dimension $d$ when $m \leq (d+2)b+1$. This dependence on the dimension - intractable for ML usage - derives from the requirement of staying within the *convex hull* of initial parameters for solving ACP: staying in the convex hull of initial parameters is increasingly difficult as the dimension increases. On the contrary, our definition of $\alpha$-robustness only requires the algorithms to improve the average squared distance to the target value. This relaxation of the consensus requirement allows us to prove *dimension independent* breakdown point. Linking LeBlanc et al. (2013)'s robustness criterion with algebraic connectivity is an interesting direction for future work.

## 4 CLIPPED GOSSIP +

In this section, we introduce a robust gossip scheme derived from ClippedGossip (He et al., 2022), but with a well-chosen and practical clipping threshold that makes it closely related to NNA (Farhadkhani et al., 2023). This scheme verifies two key properties: **(i)** a contraction property (bias-variance tradeoff); and **(ii)** it has an optimal breakdown point up to a multiplicative factor of 2. Our one-step bias-variance characterizations can directly be plugged in results from Farhadkhani et al. (2023) to show state-of-the-art guarantees for D-SGD on top of $CG^+$.

### 4.1 THE ALGORITHM

We first introduce a general update rule for Robust Gossip Aggregation (RGA) as follows:

$$\text{RGA}_{\tau_i}\left(\boldsymbol{x}_i; (\boldsymbol{x}_j)_{j \in n(i)}\right) := \boldsymbol{x}_i + \eta \sum_{j \in n(i)} w_{ij} \text{Rob}(\boldsymbol{x}_j - \boldsymbol{x}_i; \tau_i), \qquad \text{(RGA)}$$

where $\text{Rob}$ is some robustification operator that preprocesses a vector using a threshold $\tau_i$, so that choosing different robustification operators and thresholds will lead to different algorithms. We now show that ClippedGossip and NNA follow this framework.

**Making Gossip Robust.** ClippedGossip (He et al., 2022) is a robust aggregation algorithm, in which each node projects the parameters declared by its neighbors on a ball centered at its own parameter before performing a local averaging step. Given a "communication step-size" $\eta > 0$, it corresponds to Equation (RGA) where the robust aggregator is chosen as the clipping operator $\text{Rob}(v; \tau) = \text{Clip}(v; \tau) = \tau v / \max(\tau, \|v\|)$. The clipping threshold $\tau$ should be set with extreme care, as it is instrumental in limiting malicious nodes' influence, but can also slow the algorithm down, or introduce bias. With a constant clipping threshold, a constant bias is added at each step, and so one needs to limit the number of aggregation step to avoid infinite drift. A better approach is to choose the clipping threshold adaptively, depending on the pairwise distances with neighbors.

This is what He et al. (2022) do, introducing the following adaptive threshold: $\tau_i^t := \left(\frac{1}{b} \sum_{j \in n_{\mathcal{H}}(i)} w_{ij} \mathbb{E} \|\boldsymbol{x}_i^t - \boldsymbol{x}_j^t\|_2^2\right)^{1/2}$. Yet, their analysis leads to far from optimal robustness guarantees (with a breakdown least $O(\gamma)$ from the optimal one), and cannot be computed in reasonable practical

settings: not only does each node need to know the variance of the noise of its neighbors, but *it also requires to know which nodes are honest*. This breaks the fundamental assumption of not knowing the identity (honest or Byzantine) of the nodes. While an efficient rule of thumb is proposed to circumvent this, it is not supported by theory. We insist on the fact that theoretical guarantees are extremely important for robustness: resisting well against known attacks on specific datasets is not guarantee the security of a system.

**Nearest Neighbors Averaging (NNA).** Another baseline for robust decentralized averaging is NNA (Farhadkhani et al., 2023), proposed for *fully-connected communication graphs*. In this rule, each node gathers the $n(i)$ parameters from its neighbors, drops the $b$ furthest from its own, and averages the $n(i) - b$ remaining ones (trimmed mean). While Farhadkhani et al. (2023) introduce and analyze this rule for the fully-connected setting, we extend it to sparse graphs. Indeed, RGA with $\eta = 1/(m - b)$, $w_{ij} = 1$ and $\text{Rob}(v; \tau) = \text{Trim}(v; \tau) = v\mathbf{1}(\|v\| \leq \tau)$ where $\mathbf{1}$ is the indicator function and $\tau$ is chosen as the $b + 1$-th largest $\|x_i - x_j\|$ exactly recovers NNA. *The largest updates are dropped instead of clipped.* In the remainder of this paper, we refer to RGA with the trimming operator and this specific $\tau$ (but with arbitrary $\eta$ and $w_{ij}$) as NNA, thus omitting the fact that we actually refer to our sparse graphs extension. We give an analysis for this version of NNA in Theorem 3. Note that in Farhadkhani et al. (2023), a subset of the nodes is allowed not to respond to account for messages loss, or Byzantine nodes deciding not to send messages. We do not consider such a variation here.

$\text{CG}^+$**: Best of both worlds.** We now go back to the general robust aggregation framework, and introduce the following clipping rule: we choose the clipping threshold $\tau_{i,t}$ as the largest value such that the sum of the weights of the edges that are greater than this threshold is larger than $2b$, i.e.,

$$\tau_{i,t}^{\text{CG}^+} = \max\left\{\tau, \sum_{j \in n(i),\ \|\boldsymbol{x}_i^t - \boldsymbol{x}_j^t\| \geq \tau} w_{ij} \geq 2b\right\}. \tag{3}$$

If all edge weights are equal ($w_{ij} = 1$), this corresponds to taking the $2b$-th largest value of the set of edge differences. This threshold can then be used in (RGA) with clipping as a robust aggregator to obtain $\text{CG}^+$. Note that this threshold has similarities to the one discussed for NNA, using the local number of neighbors $n(i)$ instead of $m$ and taking $2b$ instead of $b + 1$, but $\text{CG}^+$ uses the clipping instead trimming. Therefore, $\text{CG}^+$ is an interesting midpoint between ClippedGossip and NNA: it can be viewed either as Gossip-NNA but with the clipping operator, or performing ClippedGossip but with NNA-type relative thresholds.

Here, $b$ is a parameter of the algorithm, which corresponds to the number of Byzantine nodes (or their total edge weight) that we would like to be robust to. We do not need to know the exact number of Byzantine nodes, but simply need to specify to how many we would like to be robust.

## 4.2 CONVERGENCE RESULTS FOR $\text{CG}^+$

As briefly discussed in the introduction, **the goal of communicating is to reduce the variance, which comes at the price of bias**. This is unavoidable, since communicating allows nodes to inject wrong information which biases the system. We now tightly quantify how much a single step of $\text{CG}^+$ reduces the variance, and how much bias is injected in the process. Let us denote $\text{Var}_{\mathcal{H}}(\boldsymbol{x}) = \frac{1}{|\mathcal{H}|} \sum_{i \in \mathcal{H}} \|\boldsymbol{x}_i - \overline{\boldsymbol{x}}_{\mathcal{H}}\|^2$, the variance of honest nodes.

**Theorem 2.** *Let $b$ and $\mu_{\min}$ be such that $4b \leq \mu_{\min}$, and let $\mathcal{G} \in \Gamma_{\mu_{\min}, b}$. Then, assuming $\eta \leq \mu_{\max}(\mathcal{G}_{\mathcal{H}})^{-1}$, the output $\boldsymbol{y} = \text{CG}^+(\boldsymbol{x})$ (obtained by one step of $\text{CG}^+$ on $\mathcal{G}$ from $\boldsymbol{x}$) verifies:*

$$\frac{1}{|\mathcal{H}|} \sum_{i \in \mathcal{H}} \|\boldsymbol{y}_i - \overline{\boldsymbol{x}}_{\mathcal{H}}\|^2 \leq (1 - \eta(\mu_{\min} - 4b)) \text{Var}_{\mathcal{H}}(\boldsymbol{x}) \tag{4}$$

$$\|\overline{\boldsymbol{y}}_{\mathcal{H}} - \overline{\boldsymbol{x}}_{\mathcal{H}}\|^2 \leq 4\eta b \text{Var}_{\mathcal{H}}(\boldsymbol{x}). \tag{5}$$

*In particular, $\text{CG}^+$ is $(1 - \eta(\mu_{\min} - 4b))$-robust on all $\mathcal{G} \in \Gamma_{\mu_{\min}, b}$. If nodes are somehow able to clip exactly $b$ honest and $b$ byzantine neighbors (which we refer to as the oracle rule), then the $4b$ factor turns into a $2b$ factor and $\text{CG}^+$ has optimal guarantees.*

Note that Equation (4) cannot be chained directly, but it can if we notice that $\text{Var}_{\mathcal{H}}(y) \leq \frac{1}{|\mathcal{H}|} \sum_{i \in \mathcal{H}} \|\boldsymbol{y}_i - \overline{\boldsymbol{x}}_{\mathcal{H}}\|^2$. We refer the reader to Appendix C for the proof. While the bound on

parameter $\eta$ depend on the honest subgraph, $\mu_{\max}(\mathcal{G}_\mathcal{H}) \leq \mu_{\max}(\mathcal{G})$, so $\eta$ can be set conservatively by evaluating $\mu_{\max}$ on the whole graph.

**Near-optimal breakdown point.** Theorem 2 shows that the upper bound on the breakdown point from Section 3 is nearly tight, since it shows that if $4b \leq \mu_{\min}$, $\text{CG}^+$ is $\alpha$-robust on $\mathcal{G}$ for any $\mathcal{G} \in \Gamma_{\mu_{\min}, b}$, while Theorem 1 says that this is impossible as soon as $2b \geq \mu_{\min}$. Note that while $\alpha$-robustness is guaranteed for the whole class, the value of $\alpha$ will depend on the actual graph within the class (through $\gamma$). The only gap left is when $\mu_{\min} \in \{2b, 4b\}$. This is a significant improvement over He et al. (2022), who obtain an equivalent result, but where where the $\mu_{\min} - 4b$ factor is essentially replaced by $\mu_{\min} - c\sqrt{b\mu_{\max}}$) (for regular graphs for instance), where $c > 0$ is a constant factor. This means that they obtain a breakdown of $b \leq c^2 \gamma \mu_{\min}$, and so they lose non-negligible constant factors as well as a full $\gamma$ factor, which rapidly shrinks with the size (and connectivity) of the graph. In other words, our guarantees are comparable when the number of Byzantine agents is small, but theirs collapse significantly before the actual breakdown point, whereas ours gracefully loosen.

**Chaining aggregation steps.** When low variance levels are required, it is necessary to perform several aggregation steps one after the other. This contrasts with the centralized setting, in which the variance can be brought to zero in one step. While the variance reduces at a linear rate, the bias accumulates as more robust aggregation steps are performed. We provide bounds for $t$ aggregation steps in the following Corollary.

**Corollary 1.** *Let $b$ and $\mu_{\min}$ be such that $4b \leq \mu_{\min}$, let $\mathcal{G} \in \Gamma_{\mu_{\min}, b}$, and denote $\delta = \frac{4b}{\mu_{\min}}$ and $\gamma = \mu_{\min}/\mu_{\max}(\mathcal{G}_\mathcal{H})$. Then, let $(\boldsymbol{x}^t)_{t \geq 0}$ be obtained from any $\boldsymbol{x}^0$ through $\boldsymbol{x}^{t+1} = \text{CG}^+(\boldsymbol{x}^t)$, with $\eta = \mu_{\max}(\mathcal{G}_\mathcal{H})^{-1}$. We have that for any $t \geq 0$,*

$$\text{Var}_\mathcal{H}(\boldsymbol{x}^t) \leq (1 - \gamma(1-\delta))^t \, \text{Var}_\mathcal{H}(\boldsymbol{x}^0), \tag{6}$$

$$\|\overline{\boldsymbol{x}}_\mathcal{H}^t - \overline{\boldsymbol{x}}_\mathcal{H}^0\| \leq \frac{\sqrt{\gamma\delta} \left(1 - [1 - \gamma(1-\delta)]^{t/2}\right)}{1 - \sqrt{1 - \gamma(1-\delta)}} \sqrt{\text{Var}_\mathcal{H}(\boldsymbol{x}^0)}. \tag{7}$$

*When $t \to \infty$, we have that $\text{Var}_\mathcal{H}(\boldsymbol{x}^t) \to 0$ (so, consensus is reached) and:*

$$\|\overline{\boldsymbol{x}}_\mathcal{H}^t - \overline{\boldsymbol{x}}_\mathcal{H}^0\|^2 \leq \frac{\gamma\delta}{(1 - \sqrt{1 - \gamma(1-\delta)})^2} \text{Var}_\mathcal{H}(\boldsymbol{x}^0) \leq \frac{4\delta}{\gamma(1-\delta)^2} \text{Var}_\mathcal{H}(\boldsymbol{x}^0). \tag{8}$$

The proof of this Corollary is given in Appendix C.3. Equation (8) shows that while one-step convergence results ensure that the total L2 error (bias plus variance) decreases, we can be in a situation in which the total L2 distance increases after several $\text{CG}^+$ steps because of bias accumulation. This happens when the factor multiplying the variance in Equation (8) is larger than 1, which essentially happens when $\gamma \ll \delta$. Yet, despite this bias, the output of the robust aggregation procedure are (arbitrarily) close to consensus, which can be desirable.

**Dependence on the parameters.** As expected, the bias increases with the amount of Byzantine corruption (through $\delta$), and decreases as the graph becomes more connected (i.e, $\gamma \to 1$). One can then use parameter $\eta$ (up to its maximum value) to control the bias-variance trade-off.

### 4.3 CONVERGENCE RESULTS FOR OTHER VARIANTS

**Gossip-NNA.** Another important method in the decentralized setting is the adaptation of NNA (Farhadkhani et al., 2023) discussed in Section 4.1, which consists in choosing the same $\tau_i$ as in $\text{CG}^+$, but *dropping* updates such as $\|x_j - x_j\| > \tau_i$ instead of clipping them to $\tau_i$. We extend our analysis to show that gossip-NNA is robust, as shown in the following result.

**Theorem 3** (Gossip-NNA breakdown.). *Gossip-NNA also verifies the guarantees of Corollary 1, where $\delta$ is replaced by $\tilde{\delta} = 8b/\mu_{\min}$.*

The proof of this result can be found in Corollary 4. When specialized to the fully-connected case (i.e., standard NNA), our bound improves on the existing one (Farhadkhani et al., 2023). Note that $\tilde{\delta} > \delta$ for $b > 1$, so NNA is worse than $\text{CG}^+$. This is not an artifact of the analysis, as we verify in Section 5, where we show experiments in which NNA breaks before $\text{CG}^+$ does.

**ClippedGossip.** We show comparable performance results for ClippedGossip, though it uses an unimplementable clipping rule, so the experimental section compares with a different rule for ClippedGossip for which no theory is established.

**Theorem 4** (Oracle ClippedGossip breakdown.). ClippedGossip *with the oracle clipping rule also verifies the guarantees of Corollary 1, where $\delta$ is replaced by $\tilde{\delta} = 8b/\mu_{\min}$.*

### 4.4 BYZANTINE ROBUST DISTRIBUTED SGD ON GRAPHS

We now give convergence results for a D-SGD-type algorithm which uses $\mathrm{CG}^+$ for decentralized robust aggregation. Several works on Byzantine-robust SGD abstract away the aggregation procedure through some contraction properties (Karimireddy et al., 2021; Wu et al., 2023; Farhadkhani et al., 2023), so that the global D-SGD result follows from the robustness of the averaging procedure. Corollary 2 builds on the reduction from Farhadkhani et al. (2023), since their requirements on the aggregation procedure exactly matches the guarantees of Theorem 2. We consider Problem 1, where we assume that each local function $f_i$ is a risk computed using a loss $\ell$ on a data distribution $\mathcal{D}_i$, i.e $f_i(\boldsymbol{x}) = \mathbb{E}_{\boldsymbol{\xi} \sim \mathcal{D}_i}[\nabla \ell(\boldsymbol{x}, \boldsymbol{\xi})]$. We propose to solve Problem 1 using decentralized stochastic gradient descent over a communication network $\mathcal{G}$. Robustness to Byzantine nodes is obtained using $\mathrm{CG}^+$ as the aggregation rule, coupled with Polyak momentum to reduce the stochastic noise.

---

**Algorithm 1** Byzantine-Resilient Decentralized SGD with $\mathrm{CG}^+$

---

**Input:** Initial model $\boldsymbol{x}_i^0 \in \mathbb{R}^d$, local loss functions $f_i$, initial momentum $m_i^0 = 0$, momentum coefficient $\beta = 0$, learning rate $\rho$, communication step size $\eta = \mu_{\max}(\mathcal{G}_{\mathcal{H}})^{-1}$, assumption on Byzantine local corruption $b$.
**for** $t = 0$ **to** $T$ **do**
    **for** $i \in \mathcal{H}$ **in parallel do**
        Compute a noisy oracle of the gradient: $\boldsymbol{g}_i^t = \nabla f_i(\boldsymbol{x}_i^t) + \boldsymbol{\xi}_i^t$.
        Update the local momentum: $\boldsymbol{m}_i^t = \beta \boldsymbol{m}_i^{t-1} + (1-\beta)\boldsymbol{g}_i^t$.
        Make an optimization step: $\boldsymbol{x}_i^{t+1/2} = \boldsymbol{x}_i^t - \rho \boldsymbol{m}_i^t$.
        Communicate parameters $\boldsymbol{x}_i^{t+1/2}$ with neighbors $n(i)$. Receive their parameters.
        Update the model using the gossip scheme: $\boldsymbol{x}_i^{t+1} = \mathrm{CG}^+\left(\boldsymbol{x}_i^{t+1/2}; \{\boldsymbol{x}_j^{t+1/2}; \ j \in n(i)\}\right)$.

---

To ensure the convergence of this algorithm, we make the following standard assumptions.

**Assumption 1.** *Objective functions regularity.*

    *1. **(Smoothness)** There exists $L \geq 0$, s.t. $\forall \boldsymbol{x}, \boldsymbol{y} \in \mathbb{R}^d$, $\|\nabla f_i(\boldsymbol{x}) - \nabla f_i(\boldsymbol{y})\| \leq L\|\boldsymbol{x} - \boldsymbol{y}\|$.*

    *2. **(Bounded noise)** There exists $\sigma \geq 0$ s.t. $\forall \boldsymbol{x} \in \mathbb{R}^d$, $\mathbb{E}[\|\nabla \ell(\boldsymbol{x}, \xi) - \nabla f_i(\boldsymbol{x})\|^2] \leq \sigma^2$.*

    *3. **(Heterogeneity)** There exist $\zeta \geq 0$ s.t. $\forall \boldsymbol{x} \in \mathbb{R}^d$, $\frac{1}{\mathcal{H}}\sum_{i \in \mathcal{H}} \|\nabla f_i(\boldsymbol{x}) - \nabla f_{\mathcal{H}}(\boldsymbol{x})\|^2 \leq \zeta^2$.*

Under these assumptions, we can prove the following corollary.

**Corollary 2.** *Let $b$ and $\mu_{\min}$ be such that $4b \leq \mu_{\min}$, and let $\mathcal{G} \in \Gamma_{\mu_{\min},b}$. Under Assumption 1, for all $i \in \mathcal{H}$, the iterates produced by Algorithm 1 on $\mathcal{G}$ with $\eta \leq 1/\mu_{\max}(\mathcal{G})$ and learning rate $\rho = \mathcal{O}(1/\sqrt{T})$ (depending also on problem parameters such as $L$, $\gamma$ or $\delta$), verify as $T$ increases:*

$$\sum_{t=1}^{T} \mathbb{E}\left[\|\nabla f_{\mathcal{H}}(\boldsymbol{x}_i^t)\|^2\right] \in \mathcal{O}\left(\frac{L\sigma}{\gamma(1-\delta)\sqrt{T}} + \frac{\zeta^2}{\gamma^2(1-\delta)^2}\right) \ \text{and} \ \mathrm{Var}_{\mathcal{H}}(\boldsymbol{x}^T) \in \mathcal{O}\left(\frac{1}{T}\left(1 + \frac{\zeta^2}{\sigma^2}\right)\right)$$

*If we perform $\tilde{\mathcal{O}}(\gamma^{-1}(1-\delta)^{-1})$ steps of $\mathrm{CG}^+$ between each gradient computation, we obtain:*

$$\sum_{t=1}^{T} \mathbb{E}\left[\|\nabla f_{\mathcal{H}}(\boldsymbol{x}_i^t)\|^2\right] \in \mathcal{O}\left(\frac{L\sigma}{\sqrt{T}}\sqrt{\frac{1}{|\mathcal{H}|} + \frac{\delta}{\gamma(1-\delta)^2}} + \frac{\delta\zeta^2}{\gamma(1-\delta)}\right).$$

As shown above, the guarantees improve when performing more aggregation steps between gradients computations. Yet, the communication cost also increases significantly in that case. This corollary is obtained by combining our Theorem 2 with Theorem 1 of Farhadkhani et al. (2023), which only requires that the robust aggregation satisfies an $(\alpha, \lambda)$-*reduction property*. Our Theorem 2 ensures that $\mathrm{CG}^+$ satisfies it with $\alpha = 1 - \gamma(1-\delta)$ and $\lambda = \gamma\delta$. The multiple communication steps case corresponds to $\alpha \approx 0$ and $\lambda = 4\delta/[\gamma(1-\delta)^2]$. A detailed proof can be found in Appendix C.5.

## 5 Designing Decentralized Attacks.

Similarly to robust aggregation methods, most proposed distributed attacks focus on centralized communication networks. For instance, Baruch et al. (2019) and Xie et al. (2020) propose that all Byzantine units send the same vector to the server to poison the update. However, the decentralized setting offers an additional surface for attacks: all nodes have different parameters, and Byzantine nodes can leverage this heterogeneity to disrupt learning even further. We show how to leverage this in two different ways, and then compare robust algorithms against these attacks.

**Changing the center of the attack** Farhadkhani et al. (2023) implement existing centralized attacks in a decentralized setting by making all Byzantine nodes declare the same attack vector to their neighbors, namely, $\overline{\boldsymbol{x}}_{\mathcal{H}}^t + a$ where $a$ is the attack direction. We make Byzantine nodes declare the parameter $\boldsymbol{x}_i^t + a$ to node $i$ instead. Hence, Byzantine nodes take the parameter of the honest node they attack as the reference point for the attack instead of the average of parameters. This reduces the likelihood of updates being clipped, but still pushes the overall system in the same direction.

**Topology-Aware Attack.** We now introduce the *Spectral Heterogeneity* (SP) attack, specifically designed to drive honest nodes away farther apart. To this end, we leverage the matrix formulation of the gossip communication, thus denoting by $\boldsymbol{X}_{\mathcal{H}} = (\boldsymbol{x}_1, \ldots, \boldsymbol{x}_{|\mathcal{H}|})^T \in \mathbb{R}^{|\mathcal{H}| \times d}$ the matrix of honest parameters. To design attacks on gossip-based robust aggregation mechanisms, we model communication as a perturbation of a gossip scheme, coherently with the analysis of RGA:

$$\boldsymbol{X}_{\mathcal{H}}^{t+1} = (\boldsymbol{I}_{\mathcal{H}} - \eta \boldsymbol{W}_{\mathcal{H}}) \boldsymbol{X}_{\mathcal{H}}^t + \eta \boldsymbol{E}^t. \tag{9}$$

By doing so, we omit the impact of the robust aggregation rule, and only consider the error term due to Byzantine nodes, i.e, we assume that $[\boldsymbol{E}^t]_i = \zeta_i^t \boldsymbol{a}_i^t$ for any honest node i, where $\zeta_i^t$ is a scaling factor of the attack, and $\boldsymbol{a}_i^t$ is the direction of attack on node $i$. We will see that this leads to powerful attacks even when taking the defense mechanism into account.

**Dissensus Attack.** To disrupt the aggregation procedure, Byzantine agents might aim at maximizing the variance of the honest parameters. A natural notion of variance in a decentralized setting is the average of pairwise differences of the neighbors parameters, which corresponds to $\|\boldsymbol{X}_{\mathcal{H}}\|_{\boldsymbol{W}_{\mathcal{H}}}^2$. Finding $\boldsymbol{a}_i^t$ such that these pairwise differences are maximized at $t + 1$ writes

$$\underset{[\boldsymbol{E}^t]_i = \zeta_i^t \boldsymbol{a}_i^t}{\arg \max} \|(\boldsymbol{I}_{\mathcal{H}} - \eta \boldsymbol{W}_{\mathcal{H}}) \boldsymbol{X}_{\mathcal{H}}^t + \eta \boldsymbol{E}^t\|_{\boldsymbol{W}_{\mathcal{H}}}^2 = \underset{[\boldsymbol{E}^t]_i = \zeta_i^t \boldsymbol{a}_i^t}{\arg \max} 2\eta \langle \boldsymbol{W}_{\mathcal{H}} \boldsymbol{X}_{\mathcal{H}}^t, \boldsymbol{E}^t \rangle + o(\eta^2).$$

Hence, maximizing the heterogeneity at time $t + 1$ suggests to take $\boldsymbol{a}_i^t = [\boldsymbol{W}_{\mathcal{H}}^t \boldsymbol{X}_{\mathcal{H}}^t]_i = \sum_{j \in n_{\mathcal{H}}(i)} (\boldsymbol{x}_i^t - \boldsymbol{x}_j^t)$. This choice of $\boldsymbol{a}_i^t$ corresponds to the *Dissensus* attack proposed in He et al. (2022). However, as gossip communication is usually operated for a large number a communication rounds, maximizing only the pairwise differences at the next step is a short-sighted approach.

**Spectral Heterogeneity Attack.** Byzantines can take into account that several communication rounds are performed over iterations. This leads, at any time $t$, to maximizing for any $s \geq 0$ the pairwise differences at time $t + s$, i.e, finding

$$\arg \max_{[\boldsymbol{E}^t]_i = \zeta_i^t \boldsymbol{a}_i^t} 2\eta \langle \boldsymbol{W}_{\mathcal{H}} (\boldsymbol{I}_{\mathcal{H}} - \eta \boldsymbol{W}_{\mathcal{H}})^{2s+1} \boldsymbol{X}_{\mathcal{H}}^t, \boldsymbol{E}^t \rangle + o(\eta^2).$$

Considering the asymptotic $s \to +\infty$ leads to approximating $\boldsymbol{W}_{\mathcal{H}} (\boldsymbol{I}_{\mathcal{H}} - \eta \boldsymbol{W}_{\mathcal{H}})^{2s}$ as a projection on its eigenspace associated with the largest eigenvalue of $\boldsymbol{W}_{\mathcal{H}} (\boldsymbol{I}_{\mathcal{H}} - \eta \boldsymbol{W}_{\mathcal{H}})^{2s}$. This eigenspace corresponds to the space spanned by the eigenvector of $\boldsymbol{W}_{\mathcal{H}}$ associated with the smallest non-zero eigenvalue of $\boldsymbol{W}_{\mathcal{H}}$, i.e $\mu_{\min}$. This eigenvector (denoted $\boldsymbol{e}_{fied}$) is commonly referred to as the Fiedler vector of the graph. Its coordinates essentially sort the nodes of the graph with the two farthest nodes associated with the largest and smallest value. Hence the signs of the values in the Fiedler vector are typically used to partition the graph into two components. Our **Spectral Heterogeneity** attack consists in taking $\boldsymbol{a}_i^t = [\boldsymbol{e}_{fied} \boldsymbol{e}_{fied}^T \boldsymbol{X}_{\mathcal{H}}^t]_i$, which essentially leads Byzantine nodes to cut the graph into two by pushing honest nodes in either plus or minus $\boldsymbol{e}_{fied}^T \boldsymbol{X}_{\mathcal{H}}^t$.

**Experimental evaluation.** Our experimental setting is the same as Farhadkhani et al. (2023), from which we used the implementation. We propose here experiments on classification taks on the MNIST datastet, and we refer to Appendix A.1 for experiments on the CIFAR-10 datasets and on an

approximate averaging task.

For all experiments, we consider honest nodes to be connected using the graph of the upper bound Section 3(see Figure 5), with 26 honest nodes distributed in two cliques of 13 nodes, each honest nodes being connected to exactly 8 nodes in the other honest clique. Following Theorem 1, the breakdown point in this graph is at most $b = 8$. The gossip matrix used is the un-normalized Laplacian matrix. Similarly to Farhadkhani et al. (2023), heterogeneity is simulated by sampling data using a Dirichlet distribution (with $alpha = 5$). The main differences with Farhadkhani et al. (2023) are (i) NNA is implemented as a gossip algorithm, (ii) ClippedGossip is implemented using the adaptive rule of clipping instead of fixed thresholds, (iii) center of Byzantine attacks is chosen as describen above, (v) Dissensus and Spectral Heterogeneity attacks are implemented. See Appendix A.1 for more experimental details.

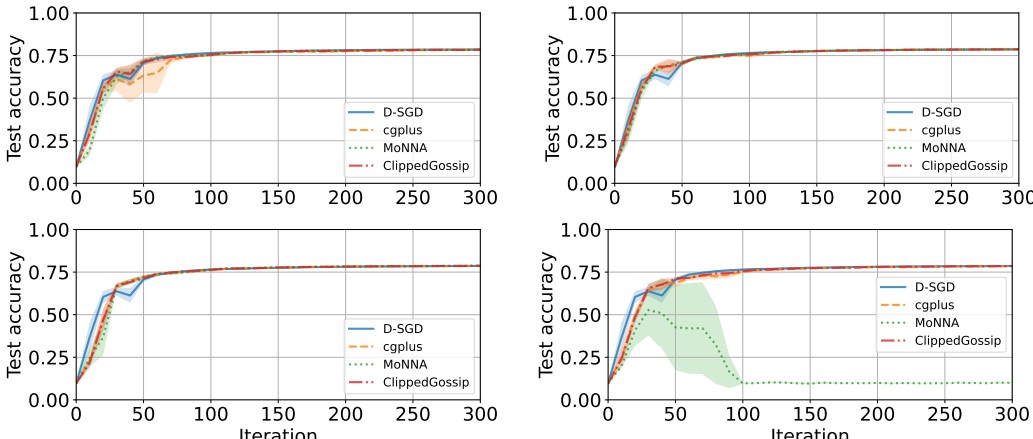

Figure 1: Accuracies of NNA, $CG^+$ and ClippedGossip on MNIST for a graph of 26 honest nodes with $\mu_{min} = 16$ in presence of $b = 6$ Byzantine neighbors to any honest node. Byzantine nodes execute *ALIE* (row 1 left), *FOE* (row 1 right), *Dissensus* (row 2 left) and *Spectral Heterogeneity* (row 2 right). D-SGD is used as a reference and is thus not attacked by Byzantine nodes.

We observe that all robust aggregation rules are robust against attacks that do not exploit the graph topology. Clipping-based attacks initially struggle against ALIE, but eventually converge to the right value. On the other hand, MoNNA fails to learn against our Spectral Heterogeneity attack, demonstrating both the efficiency of the attack and that it has a worse breakdown point than $CG^+$ (which obtains an optimal one), and so fails quicker when approaching the max theoretical breakdown point. While ClippedGossip performs on par with $CG^+$ overall, we insist on the fact that we used their rule of thumb clipping rule, which is not theoretically grounded, and thus might fail against other attacks. More details (including the link to the code repository) are given in Appendix A.

## 6    CONCLUSION

This paper revisits robust averaging over sparse communication graphs. We provide an upper bound on the optimal breakdown point, and then introduce $CG^+$, a midpoint between NNA and ClippedGossip, which meets this optimal breakdown (unlike the two other). Our experiments show that NNA indeed fails before the optimal breakdown point. To obtain this result, we introduced a new *Spectral Heterogeneity* attack that exploits the graph topology for sparse graphs. Now that we have precisely quantified the impact of the topology, an interesting future direction is the precise characterization of robustness when the constraint on the number of neighbors cannot be met globally, but local convergence can be obtained by considering that honest nodes with too many Byzantine neighbors are Byzantine themselves. Conversely, this opens up questions of which nodes should an attacker corrupt to maximize its influence for a specific graph, under the light of our results.

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

# Appendices

## A  DESCRIPTION OF THE EXPERIMENTS

Our experimental setting is built on top of the code provided by Farhadkhani et al. (2023), with the following differences:

- Attacks are designed through a linear search, but the reference point taken is the parameter of the attacked node instead of the average of all parameters. The linear search aims at maximizing $\|\boldsymbol{x}_i^{t+1} - 1/|n_{\mathcal{H}}(i)| \sum_{j \in n_{\mathcal{H}}(i)} \boldsymbol{x}_j^t\|$. As a consequence, each honest node receives different messages from Byzantine nodes.
- The aggregation is performed using a gossip update in the form of Equation (RGA) with $\eta = \mu_{\max}(\mathcal{G}_{\mathcal{H}})^{-1}$ to adapt the communication to sparse communication networks.
- Instead of considering a constant clipping threshold for ClippedGossip of He et al. (2022), as done in the experiments of (Farhadkhani et al., 2023), we use the adaptive clipping rule suggested in He et al. (2022).

The code used to run the experiments in the paper can be accessed at the following link: `https://anonymous.4open.science/r/clipped_gossip_plus_ICLR/`

### A.1  DESCRIPTION OF THE ATTACKS

In our experiments, we consider the *Dissensus* and *Spectral Heterogeneity* attacks, and two other state-of-the art attacks developed for the federated SGD setting: *Fall of Empire* (FOE) from Xie et al. (2020) and *A little is enough* (ALIE) from Baruch et al. (2019). Consistently with their original setting, these attacks rely on all Byzantine nodes declaring the same parameter $\boldsymbol{a}^t$. As we suggest in Section 5, we adapt them to the decentralized setting: Byzantine nodes declare to the honest node $i$ having the parameter $\boldsymbol{x}_i^t + \zeta_i^t \boldsymbol{a}_i^t$, where $\zeta_i^t$ is the scaling of the attack and $\boldsymbol{a}_i^t$ is the direction of the attack. The scaling $\zeta_i^t$ is chosen by Byzantine nodes through a linear search. In the case of Dissensus and Spectral Heterogeneity, $\boldsymbol{a}_i^t$ is defined as described in Section 5. In the case of FOE and ALIE, $\boldsymbol{a}_i^t$ is defined as follows:

- **ALIE**. The Byzantine nodes compute the mean of the honest parameters $\overline{\boldsymbol{x}}_{\mathcal{H}}^t$ and the coordinate-wise standard deviation $\boldsymbol{\sigma}^t$. Then they declare the parameter $\boldsymbol{a}_i^t = \boldsymbol{\sigma}^t$.
- **FOE**. The Byzantine nodes declare $\boldsymbol{a}_i^t = -\overline{\boldsymbol{x}}_{\mathcal{H}}^t$.

## B  ADDITIONAL EXPERIMENTS

### B.1  MNIST DATASET

In Figure 2 we provide extensive experiments on the MNIST classification task from Section 5. Complementary to Figure 1, we provide accuracy and training loss reached after 300 optimization steps by running for each setting 5 different seeds, and we plot the average loss and accuracy, as well as the minimal and maximal value over the different seeds.

We observe that Spectral heterogeneity is the first attack making ClippedGossip and NNA break, before Dissensus and ALIE. In this setting where the classification task is quite easy, and the heterogeneity is low (as $\alpha = 5$), the connectivity of the graphs appears as the major limiting factor to robustness of algorithms, hence Spectral Heterogeneity is very efficient. Furthermore, we note that $\text{CG}^+$ and ClippedGossip have similar performances before $b = 9$

### B.2  CIFAR-10 DATASET.

We provide in Figure 3 additional experiments on CIFAR-10 dataset. Following Farhadkhani et al. (2023), we use a CNN with four convolutional layers and two fully-connected layers. Furthermore, we set $\rho = 0.5$ and $T = 2000$ iterations. We consider the same communication network as for the

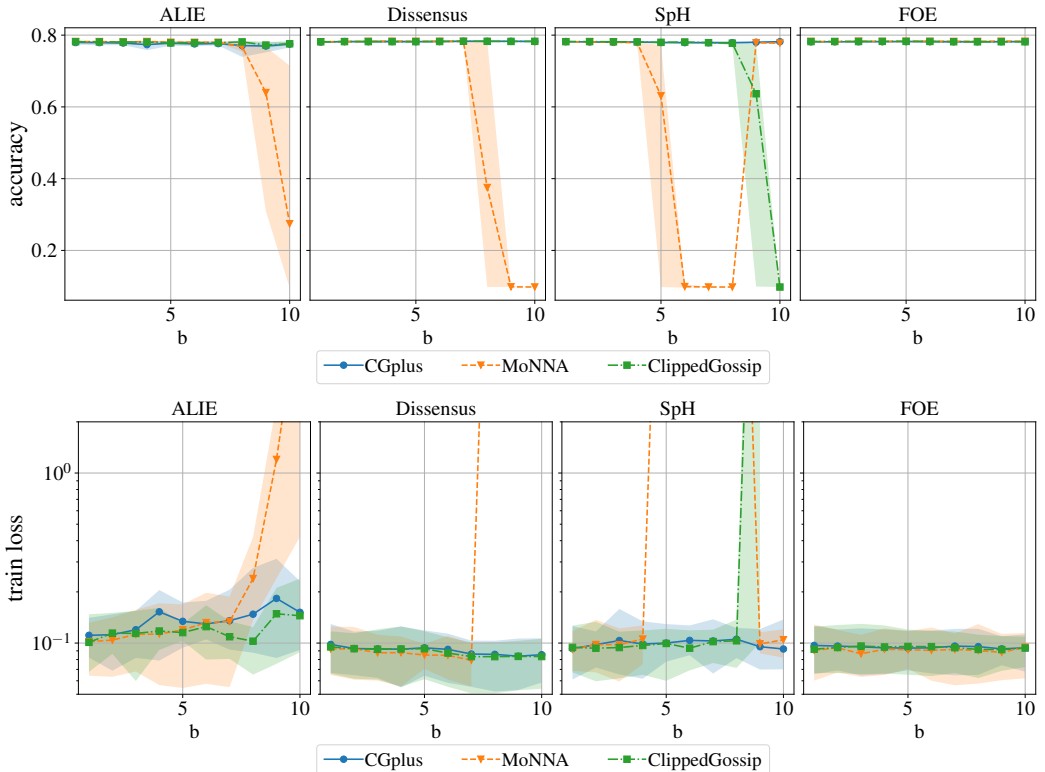

Figure 2: Test accuracies and training loss achieved by MoNNA, $CG^+$ and ClippedGossip on MNIST against 4 attacks after 300 optimization and communication steps. There are $|\mathcal{H}| = 26$ honest workers, each is neighbor to a varying number $b \in \{1, \ldots, 10\}$ of Byzantine nodes. The communication graph consists of two fully connected cliques of 13 honest nodes, each honest node is connected to 8 nodes in the other clique, hence $\mu_2(\mathcal{G}_{\mathcal{H}}) = 16$.

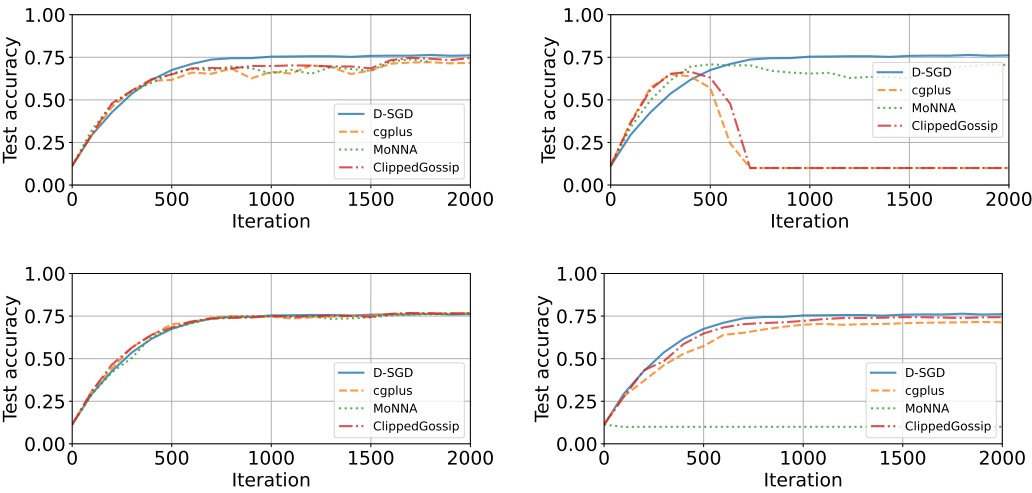

Figure 3: Test accuracies achieved by D-SGD, MoNNA, $CG^+$ and ClippedGossip on CIFAR-10 against 4 attacks, namely *ALIE* (row 1 left), *FOE* (row 1 right), *Dissensus* (row 2 left) and *Spectral Heterogeneity* (row 2 right). D-SGD is used as a reference and is thus not attacked by Byzantine nodes. There are $|\mathcal{H}| = 26$ honest workers, each is neighbor to $b = 5$ Byzantine nodes and $\mu_2(\mathcal{G}) = 16$. The communication graph consists of two fully connected cliques of 13 honest nodes, each honest node is connected to 8 nodes in the other clique.

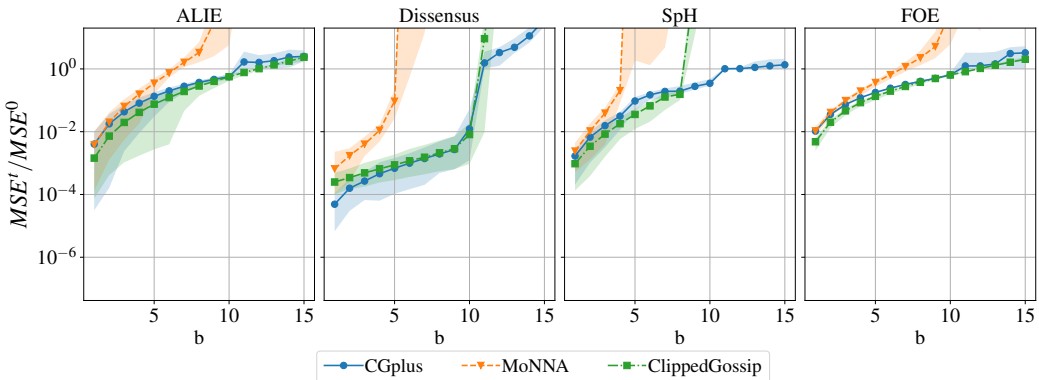

Figure 4: Relative mean square error of NNA, $CG^+$ and ClippedGossip on an averaging task. Here the optimal breakdown point is $b = 8$.

MNIST topology, with $b = 5$ Byzantine nodes neighbor to each honest node, and $\mu_2(\mathcal{G}_{\mathcal{H}}) = 16$. Up to now, we performed experiments using one seed only due to experimental running time, hence to relative performance of the different methods is to be taken with precautions.

We notice on the experiments that NNA is non-robust in this setting to the Spectral Heterogeneity attack, while ClippedGossip and $CG^+$ break with FOE attack.

### B.3 AVERAGING TASK

To finely compare the different communication schemes, we provide in Figure 4 further experiments on a the simpler task of computing the average of the honest parameters values. We consider the previous graph based on two fully-connected cliques of 13 honest nodes. Honest nodes parameters are initialized with a $\mathcal{N}(\mu, I_d/d)$ distribution ($d = 5$), where $\mu$ is equal to $+(5, 0, \ldots, 0)^T$ for one of the two clique, and equal to $-(5, 0, \ldots, 0)^T$ for the other one. As previously, each honest node in one clique is connected to 8 honest nodes in the other clique. We perform only communication steps using $CG^+$, ClippedGossip with the adaptive clipping rule, and the gossip version of NNA, and test these under the ALIE, FOE, Dissensus and Spectral Heterogeneity attacks. Eventually we plot the relative mean-square error $\sum_{i \in \mathcal{H}} \|x_i^t - \overline{x}_{\mathcal{H}}^0\|^2 / \sum_{i \in \mathcal{H}} \|x_i^0 - \overline{x}_{\mathcal{H}}^0\|^2$ after $t = 100$ communication steps. For each data point, we sample 10 different initializations, and we plot the average result, as well as the minimal and maximal value.

## C PROOFS

### C.1 PROOF OF THEOREM 1 - UPPER BOUND ON THE BREAKDOWN POINT.

Let $\mu_{\min}, b$ be such that $\mu_{\min} \leq 2b$. Let $H$ be an even number larger than $2b$.

To prove Theorem 1, we consider a communication network $G_{H,b}$ composed of three cliques of $m = |\mathcal{H}|/2$ nodes $C_1$, $C_2$ and $C_3$. Each node in $C_i$ is additionally connected to exactly $b$ nodes in $C_{i+1 \mod 3}$ and to $b$ nodes in $C_{i-1 \mod 3}$. Moreover, those connections are assumed to be *in circular order*, i.e., for any $j \in [m]$, node $j$ in $C_i$ is connected to nodes $j, \ldots, j + b \mod m$ in $C_{i+1 \mod 3}$ and $C_{i-1 \mod 3}$. If we assume that honest nodes can have up to $b$ Byzantine neighbors, then any of the three cliques can be composed of Byzantine adversaries.

The proof then goes as follows: we first show by contradiction that no $\alpha$-robust algorithm is possible in this setting, and then that $b = 2\mu_{\min}$ for this specific graph, so that $G_{H,b} \in \Gamma_{\mu_{\min},b}$. To show the contradiction, we first assume that there exists an $\alpha$-robust algorithm on $G_{H,b}$ and then:

- We show that if all nodes within one clique hold the same parameter $x^t$, and receive this parameter from nodes of either of the two other cliques, then they cannot change their parameter.

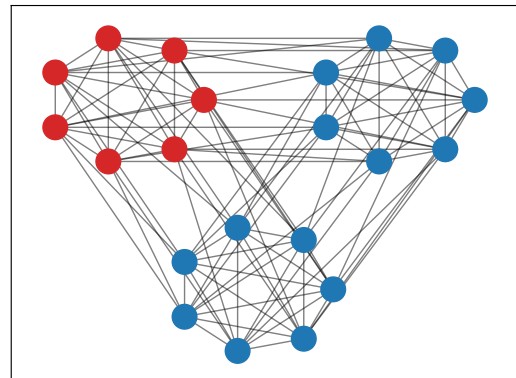

Figure 5: Topology of $G_{H,b}$ in the lower bound: two cliques are honest, one is Byzantine.

- We consider a setting where the two honest cliques holds different parameters, and we conclude that Byzantine nodes can force all honest nodes to keep their initial parameter at all times. This shows that in the considered setting, $\alpha < 1$ is impossible.

### C.1.1  No algorithm can be $\alpha$-robust on $\mathcal{G}_{\mathcal{H},b}$.

**Lemma 1.** *Consider three clique $C_1$, $C_2$ and $C_3$ of $m$ nodes. And say $\mathcal{G}$ is the graph composed by these three cliques, and that any node within one of the three cliques is connected to exactly $b \leq m$ nodes in each of the other two cliques. Assume one of these cliques is made of Byzantine nodes, then no communication algorithm is $\alpha$-robust on $G_{H,b}$.*

*Proof.*

**Part I.** Assume that there exists an algorithm $A$ that is $\alpha$-robust on $G_{h,b}$. We denote $\hat{\boldsymbol{x}}_i$ the output from node $i$ after running $A$. We consider the following setting: nodes in one clique, say $C_1$, are honest and hold the same parameter $\boldsymbol{x}^0$. Nodes in another clique, say $C_2$, declare the parameter $\boldsymbol{x}^0$ as well, while nodes in $C_3$ declare another parameter. We show that all nodes $i \in C_1$ must output the parameter $\hat{\boldsymbol{x}}_i = \boldsymbol{x}$.

As a matter of fact, from the point of view of nodes in $C_1$, it is impossible to distinguish between these two settings:

- Setting I: $C_2$ is honest, and $C_3$ is Byzantine.

- Setting II: $C_2$ is Byzantine, and $C_3$ is honest.

Consequently, nodes in $C_1$ act in the same way in both settings. Furthermore, in Setting I, nodes of $C_2$ are honest, and nodes in $C_1$ and $C_2$ have the initial same parameter; hence, the initial error is $0$. Yet the $\alpha$ criterion writes

$$\sum_{i \in \mathcal{H}} \|\hat{\boldsymbol{x}}_i - \overline{\boldsymbol{x}}_{\mathcal{H}}\|^2 \leq \alpha \sum_{i \in \mathcal{H}} \|\boldsymbol{x}_i - \overline{\boldsymbol{x}}_{\mathcal{H}}\|^2 = 0.$$

It follows that for any node $i$ in $C_1$, $\hat{\boldsymbol{x}}_i = \boldsymbol{x}$, *i.e.*, nodes do not change their parameters.

**Part II.** Consider the setting where $C_1$ and $C_2$ are honest, while $C_3$ is Byzantine, and that nodes $C_1$ hold the parameter $\boldsymbol{x}$, while node in $C_2$ hold the parameter $\boldsymbol{y} \neq \boldsymbol{x}$.

As Byzantine nodes can declare different values to their different neighbors, nodes in $C_3$ can declare to nodes in $C_1$ that they hold the value $\boldsymbol{x}$, and to nodes in $C_2$ that they hold the value $\boldsymbol{y}$. Following

Part I, nodes in $C_1$ and in $C_2$ cannot update their parameter, $(\hat{\boldsymbol{x}}_i = \boldsymbol{x}_i)$. In particular:

$$\sum_{i \in \mathcal{H}} \|\hat{\boldsymbol{x}}_i - \overline{\boldsymbol{x}}_{\mathcal{H}}\|^2 \leq \alpha \sum_{i \in \mathcal{H}} \|\boldsymbol{x}_i - \overline{\boldsymbol{x}}_{\mathcal{H}}\|^2 = \alpha \sum_{i \in \mathcal{H}} \|\hat{\boldsymbol{x}}_i - \overline{\boldsymbol{x}}_{\mathcal{H}}\|^2,$$

i.e $\alpha \geq 1$ since $\sum_{i \in \mathcal{H}} \|\hat{\boldsymbol{x}}_i - \overline{\boldsymbol{x}}_{\mathcal{H}}\|^2 > 0$, which means that Algorithm $A$ is not $\alpha$-robust on $G_{H,b}$.

$\square$

### C.1.2 Eigenvalues of the considered graph.

To conclude our lower bound, we only need to show that on the considered graph, the smallest non-zero eigenvalue of the honest subgraph is equal to $2b$. This corresponds to the following lemma.

**Lemma 2.** *Let $\mathcal{G}$ be a graph defined as two cliques $C_1$ and $C_2$ of $m$ nodes, with connections between $C_1$ and $C_2$ such that any node in $C_1$ is connected to exactly $0 \leq b \leq m$ nodes in $C_2$. Then the smallest non zero eigenvalue of the Laplacian matrix of $\mathcal{G}$ is equal to $\mu_2(\mathcal{G}_{\mathcal{H}}) = 2b$.*

*Proof.* Let $M$ be a circulant matrix defined as $M = \sum_{q=0}^{b-1} J^q$, where $J$ denotes the permutation

$$J := \begin{pmatrix} 0 & 1 & 0 & \dots & 0 \\ 0 & 0 & 1 & \dots & 0 \\ \vdots & & & \ddots & \vdots \\ 0 & & & & 1 \\ 1 & 0 & 0 & \dots & 0 \end{pmatrix}.$$

The Laplacian matrix of $\mathcal{G}$ can be written as:

$$W_{\mathcal{G}} = \begin{pmatrix} mI_m - \mathbf{1}_m\mathbf{1}_m^T & 0 \\ 0 & mI_m - \mathbf{1}_m\mathbf{1}_m^T \end{pmatrix} + \begin{pmatrix} bI_m & -M \\ -M^T & bI_m \end{pmatrix}$$

Hence

$$W_{\mathcal{G}} = (b+m)I_{2m} - \begin{pmatrix} -\mathbf{1}_m\mathbf{1}_m^T & 0 \\ 0 & -\mathbf{1}_m\mathbf{1}_m^T \end{pmatrix} - \begin{pmatrix} 0 & M \\ M^T & 0 \end{pmatrix}. \tag{10}$$

This matrix decomposition allows to have the eigenvalues of the the matrix $W_{\mathcal{G}}$.

**Lemma 3.** *The eigenvalues of $W_{\mathcal{G}}$ are $\{0, 2b\} \cup \{b + m \pm |\sum_{q=0}^{b-1} \omega^{pq}|; p \in \{1, \dots, m-1\}\}$ where $\omega := \exp(\frac{2i\pi}{m})$.*

To prove the Lemma 3, we first need to following result.

**Lemma 4.** *If $A$ is a symmetric matrix in $\mathbb{R}^{2m \times 2m}$, which can be decomposed as $A = \begin{pmatrix} 0 & M \\ M^T & 0 \end{pmatrix}$, where $M \in \mathbb{R}^{m \times m}$ is a matrix with complex eigenvalues $\mu_0, \dots \mu_{m-1}$.*

*Then the eigenvalues of $A$ are $\{\pm|\mu_q|; q = 0 \dots m-1\}$.*

*Proof of Lemma 4.* NB: In this specific proof, we denote by $\overline{D}$ the matrix of complex conjugate of elements in $D$.

Lemma 4 follows from

$$\begin{pmatrix} 0 & M \\ M^T & 0 \end{pmatrix} = \begin{pmatrix} 0 & U^*DU \\ U^TD\overline{U} & 0 \end{pmatrix} \underset{\overline{A}=A}{=} \begin{pmatrix} 0 & U^*DU \\ U^*\overline{D}U & 0 \end{pmatrix}.$$

Hence

$$A = \begin{pmatrix} U^* & 0 \\ 0 & U^* \end{pmatrix} \begin{pmatrix} 0 & D \\ \overline{D} & 0 \end{pmatrix} \begin{pmatrix} U & 0 \\ 0 & U \end{pmatrix}.$$

A simple calculus (using that $D$ is diagonal) yields that all eigenvalues of $\begin{pmatrix} 0 & D \\ \overline{D} & 0 \end{pmatrix}$ are $\{\pm|D_q|; q = 0 \dots m-1\}$.

$\square$

*Proof of Lemma 3.* We start from the decomposition of Equation (10) :

$$W_{\mathcal{G}} = (b+m)I_{2m} - \begin{pmatrix} -\mathbf{1}_m\mathbf{1}_m^T & 0 \\ 0 & -\mathbf{1}_m\mathbf{1}_m^T \end{pmatrix} - \begin{pmatrix} 0 & M \\ M^T & 0 \end{pmatrix}.$$

We first notice that $(\mathbf{1}_m^T, \mathbf{1}_m^T)^T$ and $(\mathbf{1}_m^T, -\mathbf{1}_m^T)^T$ are the only two eigenvectors of $\begin{pmatrix} -\mathbf{1}_m\mathbf{1}_m^T & 0 \\ 0 & -\mathbf{1}_m\mathbf{1}_m^T \end{pmatrix}$ associated with non zero eigenvalues. These are eigenvectors of $\begin{pmatrix} 0 & M \\ M^T & 0 \end{pmatrix}$ as well, and as such are eigenvector of $W_{\mathcal{G}}$ of eigenvalues $0$ and $2b$. Furthermore the three matrices of Equation (10) can be diagonalized in the same orthogonal basis.

The matrix $M$ is a circulant matrix, so it can be diagonalized in $\mathbb{C}$. The eigenvalues are $\{\mu_q = \sum_{p=0}^{b-1} \omega^{pq}; q \in \{0, \ldots, m-1\}\}$, where $\omega := \exp(\frac{2i\pi}{m})$. The eigenvector associated with $\mu_q$ is $x_q = (1, \omega^q, \ldots, \omega^{(m-1)q})^T$. As such, with $U = (x_0, \ldots, x_{m-1})$ and $D = \mathrm{Diag}(\mu_0, \ldots, \mu_{m-1})$, $M$ writes:

$$M = U^* D U.$$

Considering Lemma 4, the eigenvalue of $\begin{pmatrix} 0 & M \\ M^T & 0 \end{pmatrix}$ are $\{\pm|\mu_q|; q = 0, \ldots, m-1\}$, considering that $q = 0$ corresponds to the eigenvalues $+b$ and $-b$, hence the eigenvectors $(\mathbf{1}_m^T, \mathbf{1}_m^T)^T$ and $(\mathbf{1}_m^T, -\mathbf{1}_m^T)^T$, we deduce that the eigenvalues of $W_{\mathcal{G}}$ are $\{\pm|\mu_q|; q = 0 \ldots m-1\}$. $\qquad\square$

**End of the proof of Lemma 2**.

To prove Lemma 2, considering the decomposition of Equation (10), we only have to show that $m - b$ is always the second largest eigenvalue of the matrix

$$B := \begin{pmatrix} \mathbf{1}_m\mathbf{1}_m^T & 0 \\ 0 & \mathbf{1}_m\mathbf{1}_m^T \end{pmatrix} + \begin{pmatrix} 0 & M \\ M^T & 0 \end{pmatrix}.$$

First, considering Lemma 3, the eigenvalues of $B$ are $\{m+b, m-b\} \cup \{\pm|\mu_p|; p \in \{1, \ldots, m-1\}\}$ with $\mu_p = \sum_{q=0}^{b-1} \omega^{pq}$. As such showing that $|\mu_p| \le m - b$ if $p \in \{1, \ldots, m-1\}\}$ yields the result.

As $\omega^{mp} = \omega^{0p} = 1$, we have that $\sum_{q=0}^{m-1} \omega^{pq}(1 - \omega^p) = 0$. Hence, for $p \in \{1, \ldots, m-1\}$, as $\omega^p \neq 1$,

$$\sum_{q=0}^{m-1} \omega^{pq} = 0 \implies \mu_p = \sum_{q=0}^{b-1} \omega^{pq} = -\sum_{q=b}^{m-1} \omega^{pq}.$$

It follows from $|\omega| = 1$ that for $p \in \{1, \ldots, m-1\}$, $|\mu_p| \le m - b$. $\qquad\square$

## C.2 CONVERGENCE OF CG$^+$

Before proving the convergence of CG$^+$, let us first argue that Gossip with CG$^+$ only has a linear computational overhead. Indeed, the first step of computing $\tau_{i,t}^{\mathrm{CG}^+}$ requires that node $i$ performs $\mathcal{O}\left(d \cdot |n(i)|\right)$ computations, and the third one can be done in $\mathcal{O}\left(|n(i)|\right)$ on average using Quickselect, so that CG$^+$ only requires $\mathcal{O}\left(d \cdot |n(i)|\right)$ computations. Consequently, in average, CG$^+$ has the same linear complexity with respect to $d$ and $|n(i)|$ as simple averaging.

We now prove Theorem 2, and then use it to derive convergence for the Byzantine-robust decentralized optimization framework. And that all nodes follow the update scheme bellow.

$$\begin{cases} \boldsymbol{x}_i^{t+1} = \boldsymbol{x}_i^t + \eta \sum_{j \in n(i)} w_{ij} \mathrm{Rob}(\boldsymbol{x}_j^t - \boldsymbol{x}_i^t; \tau_i^t) & \text{if } i \in \mathcal{H} \\ \boldsymbol{x}_i^{t+1} = * & \text{if } i \in \mathcal{B}, \end{cases} \tag{11}$$

Where we denoted by $w_{ij} = -\boldsymbol{W}_{ij} \ge 0$ the weight associated with the edge $i \sim j$ on the graph.

Before proving Theorem 2, we introduce the following notations:

- The matrix of honest parameters $\boldsymbol{X}_{\mathcal{H}}^t := \begin{pmatrix} (\boldsymbol{x}_1^t)^T \\ \vdots \\ (\boldsymbol{x}_{|\mathcal{H}|}^t)^T \end{pmatrix} \in \mathbb{R}^{|\mathcal{H}| \times d}$.

- The error due to robust aggregation and Byzantine corruption:

$$\forall i \in \mathcal{H}, \quad [\boldsymbol{E}^t]_i := \sum_{j \in n_{\mathcal{H}}(i)} w_{ij} \left( \boldsymbol{x}_i^t - \boldsymbol{x}_j^t - \mathrm{Rob}(\boldsymbol{x}_i^t - \boldsymbol{x}_j^t; \tau_i^t) \right) + \sum_{j \in n_{\mathcal{B}}(i)} w_{ij} \mathrm{Rob}(\boldsymbol{x}_j^t - \boldsymbol{x}_i^t; \tau_i^t))$$

**Lemma 5.** *Equation* (11) *writes*

$$\boldsymbol{X}_{\mathcal{H}}^{t+1} = (\boldsymbol{I}_{\mathcal{H}} - \eta \boldsymbol{W}_{\mathcal{H}}) \boldsymbol{X}_{\mathcal{H}}^t + \eta \boldsymbol{E}^t.$$

*Proof.* Let $i \in \mathcal{H}$. We decompose the update due to the gossip scheme and consider the error term coming from both robust aggregation and the influence of Byzantine nodes.

$$\boldsymbol{x}_i^{t+1} = \boldsymbol{x}_i^t + \eta \sum_{j \in n(i)} w_{ij} \mathrm{Rob}(\boldsymbol{x}_j^t - \boldsymbol{x}_i^t; \tau_i^t)$$

$$= \boldsymbol{x}_i^t - \eta \sum_{j \in n_{\mathcal{H}}(i)} w_{ij} \mathrm{Rob}(\boldsymbol{x}_i^t - \boldsymbol{x}_j^t; \tau_i^t) + \eta \sum_{j \in n_{\mathcal{B}}(i)} w_{ij} \mathrm{Rob}(\boldsymbol{x}_j^t - \boldsymbol{x}_i^t; \tau_i^t)$$

$$\boldsymbol{x}_i^{t+1} = \boldsymbol{x}_i^t - \eta \sum_{j \in n_{\mathcal{H}}(i)} w_{ij} (\boldsymbol{x}_i^t - \boldsymbol{x}_j^t)$$

$$+ \eta \sum_{j \in n_{\mathcal{H}}(i)} w_{ij} \left[ (\boldsymbol{x}_i^t - \boldsymbol{x}_j^t) - \mathrm{Rob}\left(\boldsymbol{x}_i^t - \boldsymbol{x}_j^t; \tau_i^t\right) \right] + \eta \sum_{j \in n_{\mathcal{B}}(i)} w_{ij} \mathrm{Rob}(\boldsymbol{x}_j^t - \boldsymbol{x}_i^t; \tau_i^t)$$

Finally, the proof is concluded by remarking that $[W_{\mathcal{H}} \boldsymbol{X}_{\mathcal{H}}^t]_i = \sum_{j \in n_{\mathcal{H}}(i)} w_{ij} (\boldsymbol{x}_i^t - \boldsymbol{x}_j^t)$. $\qquad\square$

We begin by controlling the norm of the error term $\|\boldsymbol{E}^t\|_2^2$ in the case of $\mathrm{CG}^+$.

**Lemma 6** (Control of the error, $\mathrm{CG}^+$). *The error due to clipping and Byzantine nodes is controlled by the heterogeneity as measured by the gossip matrix:*

$$\|\boldsymbol{E}^t\|_2^2 \leq 4b \|\boldsymbol{X}_{\mathcal{H}}^t\|_{\boldsymbol{W}_{\mathcal{H}}}^2 = 2b \sum_{i \in \mathcal{H}, j \in n_{\mathcal{H}}(i)} w_{ij} \|\boldsymbol{x}_i^t - \boldsymbol{x}_j^t\|^2$$

*Proof.* We recall that in this case,

$$\forall i \in \mathcal{H}, \quad [\boldsymbol{E}^t]_i := \sum_{j \in n_{\mathcal{H}}(i)} w_{ij} \left( \boldsymbol{x}_i^t - \boldsymbol{x}_j^t - \mathrm{Clip}(\boldsymbol{x}_i^t - \boldsymbol{x}_j^t; \tau_i^t) \right) + \sum_{j \in n_{\mathcal{B}}(i)} w_{ij} \mathrm{Clip}(\boldsymbol{x}_j^t - \boldsymbol{x}_i^t; \tau_i^t))$$

By applying the triangle inequality, and by denoting $(a)_+ = \max(a, 0)$, we get

$$\|\boldsymbol{E}^t\|^2 = \sum_{i \in \mathcal{H}} \left\| \sum_{j \in n_{\mathcal{H}}(i)} w_{ij} \left( \boldsymbol{x}_i^t - \boldsymbol{x}_j^t - \mathrm{Clip}(\boldsymbol{x}_i^t - \boldsymbol{x}_j^t; \tau_i^t) \right) + \sum_{j \in n_{\mathcal{B}}(i)} w_{ij} \mathrm{Clip}(\boldsymbol{x}_i^t - \boldsymbol{x}_j^t; \tau_i^t) \right\|_2^2$$

$$\leq \sum_{i \in \mathcal{H}} \left( \sum_{j \in n_{\mathcal{H}}(i)} w_{ij} \| \boldsymbol{x}_i^t - \boldsymbol{x}_j^t - \mathrm{Clip}(\boldsymbol{x}_i^t - \boldsymbol{x}_j^t; \tau_i^t) \|_2 + \sum_{j \in n_{\mathcal{B}}(i)} w_{ij} \| \mathrm{Clip}(\boldsymbol{x}_i^t - \boldsymbol{x}_j^t; \tau_i^t) \|_2 \right)^2$$

$$\leq \sum_{i \in \mathcal{H}} \left( \sum_{j \in n_{\mathcal{H}}(i)} w_{ij} \left( \|\boldsymbol{x}_i^t - \boldsymbol{x}_j^t\|_2 - \tau_i^t \right)_+ + b\tau_i^t \right)^2,$$

where we recall that for all $i \in \mathcal{H}$, $b \geq \sum_{j \in n_{\mathcal{B}}(i)} w_{ij}$.

We define $\tau_i^t$ as

$$\tau_i^t := \max \left\{ \tau \geq 0 : \sum_{j \in n(i)} w_{ij} \mathbf{1}_{\|\boldsymbol{x}_i^t - \boldsymbol{x}_j^t\| \geq \tau} \geq 2b \right\}.$$

This corresponds to lower the clipping threshold until the sum of weight of clipped neighbors is essentially equal to $2b$. This allows to ensure that at the sum of the weights of honest neighbors of $i$ that are clipped is at least $b$ and at most $2b$. In presence of ties at the clipping threshold, honest neighbors can be arbitrarily considered as *clipped* or *non clipped* or, said differently, there is no clipping error incurred since the clipping threshold is the same as the actual value of the difference. Therefore, we do not have to accumulate error for the weight over $2b$, so that the following equation always holds:

$$2b \geq \sum_{j \in n_{\mathcal{H}}(i)} w_{ij} \mathbf{1}_{j \text{ clipped}} \geq b.$$

If somehow we have an oracle that allows us to only clip neighbors corresponding to weight equal to $b$ (which amounts to being able to pick a set of size $2b$ which we know contains all Byzantine nodes, a very strong and generally unimplementable rule) we would obtain that the weight of clipped nodes is exactly $b$, thus motivating our remark.

In both cases, the lower bound on the weights of clipped honest neighbors yeilds:

$$\sum_{j \in n_{\mathcal{H}}(i)} w_{ij} \left( \|\boldsymbol{x}_i^t - \boldsymbol{x}_j^t\|_2 - \tau_i^t \right)_+ + b\tau_i^t \leq \sum_{j \in n_{\mathcal{H}}(i)} w_{ij} (\|\boldsymbol{x}_i^t - \boldsymbol{x}_j^t\|_2 - \tau_i^t) \mathbf{1}_{j \text{ clipped}} + b\tau_i^t$$

$$\leq \sum_{j \in n_{\mathcal{H}}(i)} w_{ij} \|\boldsymbol{x}_i^t - \boldsymbol{x}_j^t\|_2 \mathbf{1}_{j \text{ clipped}}.$$

We conclude the proof using the Cauchy-Schwarz inequality:

$$\|\boldsymbol{E}^t\|^2 \leq \sum_{i \in \mathcal{H}} \left( \sum_{j \in n_{\mathcal{H}}(i)} \sqrt{w_{ij}} \|\boldsymbol{x}_i^t - \boldsymbol{x}_j^t\| \cdot \sqrt{w_{ij}} \mathbf{1}_{j \text{ clipped}} \right)^2$$

$$\leq \sum_{i \in \mathcal{H}} \left( \sum_{j \in n_{\mathcal{H}}(i)} w_{ij} \mathbf{1}_{j \text{ clipped}} \right) \left( \sum_{j \in n_{\mathcal{H}}(i)} w_{ij} \|\boldsymbol{x}_i^t - \boldsymbol{x}_j^t\|^2 \right)$$

$$\leq \sum_{i \in \mathcal{H}} (2b + \max_{j \in n_{\mathcal{H}}(i)} w_{ij}) \sum_{j \in n_{\mathcal{H}}(i)} w_{ij} \|\boldsymbol{x}_i^t - \boldsymbol{x}_j^t\|^2$$

$$\|\boldsymbol{E}^t\|^2 \leq 2b \sum_{i \in \mathcal{H}} \sum_{j \in n_{\mathcal{H}}(i)} w_{ij} \|\boldsymbol{x}_i^t - \boldsymbol{x}_j^t\|^2.$$

Where we used $2b \geq \sum_{j \in n_{\mathcal{H}}(i)} w_{ij} \mathbf{1}_{j \text{ clipped}}$. Note that the factor 2 disappear if we can somehow manage to clip exactly the weight of $b$ honest nodes.

The result finally follows by noting that $2\|\boldsymbol{X}_{\mathcal{H}}^t\|_{\boldsymbol{W}_{\mathcal{H}}}^2 = \sum_{i\in\mathcal{H}, j\in n_{\mathcal{H}}(i)} w_{ij}\|\boldsymbol{x}_i^t - \boldsymbol{x}_j^t\|^2$. Indeed, considering that $\mathcal{G}_{\mathcal{H}}$ is an undirected graph, $i \in n_{\mathcal{H}}(j) \iff j \in n_{\mathcal{H}}(i)$ and we have:

$$\|\boldsymbol{X}_{\mathcal{H}}^t\|_{\boldsymbol{W}_{\mathcal{H}}}^2 = \langle \boldsymbol{X}_{\mathcal{H}}, \boldsymbol{W}_{\mathcal{H}}\boldsymbol{X}_{\mathcal{H}}\rangle$$

$$= \sum_{i\in\mathcal{H}} \left\langle \boldsymbol{x}_i^t, \sum_{j\in n_{\mathcal{H}}(i)} w_{ij}(\boldsymbol{x}_i^t - \boldsymbol{x}_j^t)\right\rangle$$

$$= \sum_{i\in\mathcal{H}} \sum_{j\in n_{\mathcal{H}}(i)} w_{ij}\left\langle \boldsymbol{x}_i^t, \boldsymbol{x}_i^t - \boldsymbol{x}_j^t\right\rangle$$

$$= \frac{1}{2}\sum_{i\in\mathcal{H}} \sum_{j\in n_{\mathcal{H}}(i)} w_{ij}\left\langle \boldsymbol{x}_i^t - \boldsymbol{x}_j^t, \boldsymbol{x}_i^t - \boldsymbol{x}_j^t\right\rangle$$

$$\|\boldsymbol{X}_{\mathcal{H}}^t\|_{\boldsymbol{W}_{\mathcal{H}}}^2 = \frac{1}{2}\sum_{i\in\mathcal{H}, j\in n_{\mathcal{H}}(i)} w_{ij}\left\|\boldsymbol{x}_i^t - \boldsymbol{x}_j^t\right\|_2^2$$

$\square$

Now that we control the error term, we can conclude the proof of Theorem 2 using standard optimization arguments. Before proving this theorem, we prove the following one, from which Corollary 1 is direct.

**Theorem 5.** *Let $b$ and $\mu_{\min}$ be such that $4b \le \mu_{\min}$, and let $\mathcal{G} \in \Gamma_{\mu_{\min}, b}$. Then, assuming $\eta \le \mu_{\max}(\mathcal{G}_{\mathcal{H}})^{-1}$, the output $\boldsymbol{y} = \mathrm{CG}^+(\boldsymbol{x})$ (obtained by one step of $\mathrm{CG}^+$ on $\mathcal{G}$ from $\boldsymbol{x}$) verifies:*

$$\frac{1}{|\mathcal{H}|}\sum_{i\in\mathcal{H}}\|\boldsymbol{x}_i^{t+1} - \overline{\boldsymbol{x}}_{\mathcal{H}}^{t+1}\|^2 \le (1 - \eta\,(\mu_{\min} - 4b))\frac{1}{|\mathcal{H}|}\sum_{i\in\mathcal{H}}\|\boldsymbol{x}_i^t - \overline{\boldsymbol{x}}_{\mathcal{H}}^t\|^2 \tag{12}$$

$$\|\overline{\boldsymbol{x}}_{\mathcal{H}}^{t+1} - \overline{\boldsymbol{x}}_{\mathcal{H}}^t\|^2 \le \eta\frac{4b}{|\mathcal{H}|}\sum_{i\in\mathcal{H}}\|\boldsymbol{x}_i^t - \overline{\boldsymbol{x}}_{\mathcal{H}}^t\|^2 \tag{13}$$

*Proof.* **Part I: Equation (13).**

Equation (13) is a direct consequence of Lemma 6. Indeed applying $\boldsymbol{P}_{\mathbf{1}_{\mathcal{H}}} := \frac{1}{|\mathcal{H}|}\mathbf{1}_{\mathcal{H}}\mathbf{1}_{\mathcal{H}}^T$ - the orthogonal projection on the kernel of $\boldsymbol{W}_{\mathcal{H}}$ - on Lemma 5 results in

$$\boldsymbol{P}_{\mathbf{1}_{\mathcal{H}}}\boldsymbol{X}_{\mathcal{H}}^{t+1} = \boldsymbol{P}_{\mathbf{1}_{\mathcal{H}}}(\boldsymbol{I}_{\mathcal{H}} - \eta\boldsymbol{W}_{\mathcal{H}})\boldsymbol{X}_{\mathcal{H}}^t + \eta\boldsymbol{P}_{\mathbf{1}_{\mathcal{H}}}\boldsymbol{E}^t = \boldsymbol{P}_{\mathbf{1}_{\mathcal{H}}}\boldsymbol{X}_{\mathcal{H}}^t + \eta\boldsymbol{P}_{\mathbf{1}_{\mathcal{H}}}\boldsymbol{E}^t.$$

Taking the norm yields

$$\|\boldsymbol{P}_{\mathbf{1}_{\mathcal{H}}}\boldsymbol{X}_{\mathcal{H}}^{t+1} - \boldsymbol{P}_{\mathbf{1}_{\mathcal{H}}}\boldsymbol{X}_{\mathcal{H}}^t\|^2 = \eta^2\|\boldsymbol{P}_{\mathbf{1}_{\mathcal{H}}}\boldsymbol{E}^t\|^2 \le \eta^2\|\boldsymbol{E}^t\|^2. \tag{14}$$

We now apply Lemma 6, and use that $\mu_{\max}(\mathcal{G}_{\mathcal{H}})$ is the largest eigenvalue of $\boldsymbol{W}_{\mathcal{H}}$. It gives

$$\|\boldsymbol{P}_{\mathbf{1}_{\mathcal{H}}}\boldsymbol{X}_{\mathcal{H}}^{t+1} - \boldsymbol{P}_{\mathbf{1}_{\mathcal{H}}}\boldsymbol{X}_{\mathcal{H}}^t\|^2 \le \eta^2 4b\|\boldsymbol{X}_{\mathcal{H}}^t\|_{\boldsymbol{W}_{\mathcal{H}}}^2$$
$$\le \mu_{\max}(\mathcal{G}_{\mathcal{H}})\eta^2 4b\|(\boldsymbol{I}_{\mathcal{H}} - \boldsymbol{P}_{\mathbf{1}_{\mathcal{H}}})\boldsymbol{X}_{\mathcal{H}}^t\|^2$$

Finally, Equation (13) derives from $[\boldsymbol{P}_{\mathbf{1}_{\mathcal{H}}}\boldsymbol{X}_{\mathcal{H}}^t]_{i\in\mathcal{H}} = [\sum_{j\in\mathcal{H}}\boldsymbol{x}_j^t]_{i\in\mathcal{H}} = [\overline{\boldsymbol{x}}_{\mathcal{H}}^t]_{i\in\mathcal{H}}$ and $\eta\mu_{\max}(\mathcal{G}_{\mathcal{H}}) \le 1$.

**Part II: Equation (12).**

To prove Equation (12), we consider the objective function $\|(\boldsymbol{I}_{\mathcal{H}} - \boldsymbol{P}_{\mathbf{1}_{\mathcal{H}}})\boldsymbol{X}^t\|^2$. We denote by $\boldsymbol{W}_{\mathcal{H}}^{\dagger}$ the Moore-Penrose pseudo inverse of $\boldsymbol{W}_{\mathcal{H}}$. We begin by applying Lemma 5.

$$\|(\boldsymbol{I}_{\mathcal{H}} - \boldsymbol{P}_{\boldsymbol{1}_{\mathcal{H}}})\boldsymbol{X}_{\mathcal{H}}^{t+1}\|^2 = \|\boldsymbol{X}_{\mathcal{H}}^t - \eta\boldsymbol{W}_{\mathcal{H}}\boldsymbol{X}_{\mathcal{H}}^t + \eta\boldsymbol{E}^t\|_{(\boldsymbol{I}_{\mathcal{H}} - \boldsymbol{P}_{\boldsymbol{1}_{\mathcal{H}}})}^2$$

$$= \|\boldsymbol{X}_{\mathcal{H}}^t\|_{(\boldsymbol{I}_{\mathcal{H}} - \boldsymbol{P}_{\boldsymbol{1}_{\mathcal{H}}})}^2 - 2\eta\left\langle \boldsymbol{X}_{\mathcal{H}}^t, \boldsymbol{W}_{\mathcal{H}}\boldsymbol{X}_{\mathcal{H}}^t - \boldsymbol{E}^t\right\rangle_{(\boldsymbol{I}_{\mathcal{H}} - \boldsymbol{P}_{\boldsymbol{1}_{\mathcal{H}}})}$$

$$+ \eta^2 \left\|\boldsymbol{W}_{\mathcal{H}}\boldsymbol{X}_{\mathcal{H}}^t - \boldsymbol{E}^t\right\|_{(\boldsymbol{I}_{\mathcal{H}} - \boldsymbol{P}_{\boldsymbol{1}_{\mathcal{H}}})}$$

$$= \|\boldsymbol{X}_{\mathcal{H}}^t\|_{(\boldsymbol{I}_{\mathcal{H}} - \boldsymbol{P}_{\boldsymbol{1}_{\mathcal{H}}})}^2 - 2\eta\left\langle \boldsymbol{X}_{\mathcal{H}}^t, \boldsymbol{X}_{\mathcal{H}}^t - \boldsymbol{W}_{\mathcal{H}}^\dagger\boldsymbol{E}^t\right\rangle_{\boldsymbol{W}_{\mathcal{H}}}$$

$$+ \eta^2 \left\|\boldsymbol{X}_{\mathcal{H}}^t - \boldsymbol{W}_{\mathcal{H}}^\dagger\boldsymbol{E}^t\right\|_{\boldsymbol{W}_{\mathcal{H}}^2}.$$

Applying $2\langle\boldsymbol{a}, \boldsymbol{b}\rangle = \|\boldsymbol{a}\|^2 + \|\boldsymbol{b}\|^2 - \|\boldsymbol{a} - \boldsymbol{b}\|^2$ leads to

$$\|\boldsymbol{X}_{\mathcal{H}}^{t+1}\|_{(\boldsymbol{I}_{\mathcal{H}} - \boldsymbol{P}_{\boldsymbol{1}_{\mathcal{H}}})}^2 - \|\boldsymbol{X}_{\mathcal{H}}^t\|_{(\boldsymbol{I}_{\mathcal{H}} - \boldsymbol{P}_{\boldsymbol{1}_{\mathcal{H}}})}^2$$

$$= -\eta\left\|\boldsymbol{X}_{\mathcal{H}}^t\right\|_{\boldsymbol{W}_{\mathcal{H}}}^2 - \eta\left\|\boldsymbol{X}_{\mathcal{H}}^t - \boldsymbol{W}_{\mathcal{H}}^\dagger\boldsymbol{E}^t\right\|_{\boldsymbol{W}_{\mathcal{H}}}^2 + \eta\left\|\boldsymbol{W}_{\mathcal{H}}^\dagger\boldsymbol{E}^t\right\|_{\boldsymbol{W}_{\mathcal{H}}}^2$$

$$+ \eta^2 \left\|\boldsymbol{X}_{\mathcal{H}}^t - \boldsymbol{W}_{\mathcal{H}}^\dagger\boldsymbol{E}^t\right\|_{\boldsymbol{W}_{\mathcal{H}}^2}$$

$$= -\eta\left\|\boldsymbol{X}_{\mathcal{H}}^t\right\|_{\boldsymbol{W}_{\mathcal{H}}}^2 + \eta\left\|\boldsymbol{E}^t\right\|_{\boldsymbol{W}_{\mathcal{H}}^\dagger}^2 \tag{15}$$

$$- \eta\left\|\boldsymbol{X}_{\mathcal{H}}^t - \boldsymbol{W}_{\mathcal{H}}^\dagger\boldsymbol{E}^t\right\|_{\boldsymbol{W}_{\mathcal{H}}}^2 + \eta^2\left\|\boldsymbol{X}_{\mathcal{H}}^t - \boldsymbol{W}_{\mathcal{H}}^\dagger\boldsymbol{E}^t\right\|_{\boldsymbol{W}_{\mathcal{H}}^2}.$$

We now apply that $\mu_{\max}(\mathcal{G}_{\mathcal{H}})$ (resp. $\mu_2(\mathcal{G}_{\mathcal{H}})$) is the largest (resp. smallest) non-zero eigenvalue of $\boldsymbol{W}_{\mathcal{H}}$.

$$\|\boldsymbol{X}_{\mathcal{H}}^{t+1}\|_{(\boldsymbol{I}_{\mathcal{H}} - \boldsymbol{P}_{\boldsymbol{1}_{\mathcal{H}}})}^2 - \|\boldsymbol{X}_{\mathcal{H}}^t\|_{(\boldsymbol{I}_{\mathcal{H}} - \boldsymbol{P}_{\boldsymbol{1}_{\mathcal{H}}})}^2$$

$$\leq -\eta\left\|\boldsymbol{X}_{\mathcal{H}}^t\right\|_{\boldsymbol{W}_{\mathcal{H}}}^2 + \eta\frac{1}{\mu_2(\mathcal{G}_{\mathcal{H}})}\left\|\boldsymbol{E}^t\right\|^2$$

$$- \eta(1 - \mu_{\max}(\mathcal{G}_{\mathcal{H}})\eta)\left\|\boldsymbol{X}_{\mathcal{H}}^t - \boldsymbol{W}_{\mathcal{H}}^\dagger\boldsymbol{E}^t\right\|_{\boldsymbol{W}_{\mathcal{H}}}^2.$$

Eventually Lemma 6 with the assumption $\eta \leq 1/\mu_{\max}(\mathcal{G}_{\mathcal{H}})$ yield the result

$$\|\boldsymbol{X}_{\mathcal{H}}^{t+1}\|_{(\boldsymbol{I}_{\mathcal{H}} - \boldsymbol{P}_{\boldsymbol{1}_{\mathcal{H}}})}^2 \leq \|\boldsymbol{X}_{\mathcal{H}}^t\|_{(\boldsymbol{I}_{\mathcal{H}} - \boldsymbol{P}_{\boldsymbol{1}_{\mathcal{H}}})}^2 - \eta\left(1 - \frac{4b}{\mu_2(\mathcal{G}_{\mathcal{H}})}\right)\|\boldsymbol{X}_{\mathcal{H}}^t\|_{\boldsymbol{W}_{\mathcal{H}}}^2$$

$$\|\boldsymbol{X}_{\mathcal{H}}^{t+1}\|_{(\boldsymbol{I}_{\mathcal{H}} - \boldsymbol{P}_{\boldsymbol{1}_{\mathcal{H}}})}^2 \leq \left(1 - \eta\mu_2(\mathcal{G}_{\mathcal{H}})\left(1 - \frac{4b}{\mu_2(\mathcal{G}_{\mathcal{H}})}\right)\right)\|\boldsymbol{X}_{\mathcal{H}}^t\|_{(\boldsymbol{I}_{\mathcal{H}} - \boldsymbol{P}_{\boldsymbol{1}_{\mathcal{H}}})}^2.$$

□

To obtain Theorem 5, we note that we can actually control the one-step variation of the MSE using $(1 - \eta(\mu_{\min} - 4b))$ only, thus strengthening the first inequality. We rewrite the first part of Theorem 2 below for completeness.

**Corollary 3.** *Let $b$ and $\mu_{\min}$ be such that $4b \leq \mu_{\min}$, and let $\mathcal{G} \in \Gamma_{\mu_{\min},b}$. Then, assuming $\eta \leq \mu_{\max}(\mathcal{G}_{\mathcal{H}})^{-1}$, the output $\boldsymbol{y} = \mathrm{CG}^+(\boldsymbol{x})$ (obtained by one step of $\mathrm{CG}^+$ on $\mathcal{G}$ from $\boldsymbol{x}$) verifies:*

$$\frac{1}{|\mathcal{H}|}\sum_{i \in \mathcal{H}}\|\boldsymbol{x}_i^{t+1} - \overline{\boldsymbol{x}}_{\mathcal{H}}^t\|^2 \leq (1 - \eta(\mu_{\min} - 4b))\frac{1}{|\mathcal{H}|}\sum_{i \in \mathcal{H}}\|\boldsymbol{x}_i^t - \overline{\boldsymbol{x}}_{\mathcal{H}}^t\|^2$$

*Proof.* We consider Equation (14) and Equation (15), which write

$$\|\boldsymbol{P}_{\boldsymbol{1}_{\mathcal{H}}}\boldsymbol{X}_{\mathcal{H}}^{t+1} - \boldsymbol{P}_{\boldsymbol{1}_{\mathcal{H}}}\boldsymbol{X}_{\mathcal{H}}^t\|^2 = \eta^2\|\boldsymbol{P}_{\boldsymbol{1}_{\mathcal{H}}}\boldsymbol{E}^t\|^2.$$

$$\|X_{\mathcal{H}}^{t+1}\|_{(I_{\mathcal{H}}-P_{1_{\mathcal{H}}})}^2 - \|X_{\mathcal{H}}^t\|_{(I_{\mathcal{H}}-P_{1_{\mathcal{H}}})}^2 \leq -\eta \left\|X_{\mathcal{H}}^t\right\|_{W_{\mathcal{H}}}^2 + \eta \left\|E^t\right\|_{W_{\mathcal{H}}^\dagger}^2.$$

It follows from the bias - variance decomposition of the MSE

$$\|X_{\mathcal{H}}^{t+1} - P_{1_{\mathcal{H}}}X_{\mathcal{H}}^t\|^2 = \|(I_{\mathcal{H}} - P_{1_{\mathcal{H}}})X_{\mathcal{H}}^{t+1}\|^2 + \|P_{1_{\mathcal{H}}}X_{\mathcal{H}}^{t+1} - P_{1_{\mathcal{H}}}X_{\mathcal{H}}^t\|^2$$

that

$$\|X_{\mathcal{H}}^{t+1} - P_{1_{\mathcal{H}}}X_{\mathcal{H}}^t\|^2 - \|X_{\mathcal{H}}^t\|_{(I_{\mathcal{H}}-P_{1_{\mathcal{H}}})}^2 \leq -\eta \left\|X_{\mathcal{H}}^t\right\|_{W_{\mathcal{H}}}^2 + \eta \left\|E^t\right\|_{W_{\mathcal{H}}^\dagger}^2 + \eta^2 \|P_{1_{\mathcal{H}}}E^t\|^2$$

$$\leq -\eta \left\|X_{\mathcal{H}}^t\right\|_{W_{\mathcal{H}}}^2 + \eta \frac{1}{\mu_2(\mathcal{G}_{\mathcal{H}})} \left\|E^t\right\|_{(I_{\mathcal{H}}-P_{1_{\mathcal{H}}})}^2 + \eta^2 \|P_{1_{\mathcal{H}}}E^t\|^2$$

As $\eta \leq \frac{1}{\mu_{\max}(\mathcal{G}_{\mathcal{H}})} \leq \frac{1}{\mu_2(\mathcal{G}_{\mathcal{H}})}$, we eventually get

$$\|X_{\mathcal{H}}^{t+1} - P_{1_{\mathcal{H}}}X_{\mathcal{H}}^t\|^2 \leq \|X_{\mathcal{H}}^t\|_{(I_{\mathcal{H}}-P_{1_{\mathcal{H}}})}^2 - \eta \left\|X_{\mathcal{H}}^t\right\|_{W_{\mathcal{H}}}^2 + \eta \frac{1}{\mu_2(\mathcal{G}_{\mathcal{H}})} \left\|E^t\right\|^2$$

$$\leq \|X_{\mathcal{H}}^t\|_{(I_{\mathcal{H}}-P_{1_{\mathcal{H}}})}^2 - \eta \left(1 - \frac{4b}{\mu_2(\mathcal{G}_{\mathcal{H}})}\right) \left\|X_{\mathcal{H}}^t\right\|_{W_{\mathcal{H}}}^2$$

$$\leq \left(1 - \eta\mu_2(\mathcal{G}_{\mathcal{H}})\left(1 - \frac{4b}{\mu_2(\mathcal{G}_{\mathcal{H}})}\right)\right) \|X_{\mathcal{H}}^t\|_{(I_{\mathcal{H}}-P_{1_{\mathcal{H}}})}^2$$

$\square$

### C.3   CONSEQUENCES

A direct consequence of the above results is Corollary 1, as we show below.

*Proof.* Using the $(\alpha, \lambda)$ reduction notations, we have:

$$\begin{cases} \alpha = 1 - \gamma\left(1 - \delta\right)) \\ \lambda = \gamma\delta \end{cases}$$

We denote here the drift increment $d_{t+1} = \|P_{1_{\mathcal{H}}}X_{\mathcal{H}}^{t+1} - P_{1_{\mathcal{H}}}X_{\mathcal{H}}^t\|$ and the variance at time $t$ as $\sigma_t^2 = \|X_{\mathcal{H}}^{t+1}\|_{(I_{\mathcal{H}}-P_{1_{\mathcal{H}}})}^2$.

Corollary 3 ensures that

$$\sigma_{t+1}^2 + d_t^2 \leq \alpha\sigma_t^2.$$

Hence,we have $\sigma_{t+1}^2 + d_{t+1}^2 \leq \alpha\sigma_t^2$, and so $\sigma_{t+1}^2 \leq \alpha\sigma_t^2$, which implies that $\sigma_t \leq \alpha^{t/2}\sigma_0$. This proves the first part of the result. Using this, we write that Theorem 5 ensures that

$$d_{t+1} \leq \sqrt{\lambda}\sigma_t \leq \sqrt{\lambda}\beta^t\sigma0,$$

leading to:

$$\sum_{t=1}^{T} d_t \leq \sqrt{\lambda}\sum_{t=0}^{T-1}\sigma_t \leq \sqrt{\lambda}\sum_{t=0}^{T-1}\alpha^{t/2}\sigma_0 \leq \frac{\sqrt{\lambda}(1-\alpha^{T/2})}{1-\alpha^{1/2}}\sigma0,$$

which proves the second part. The last inequality is obtained by writing.

$$\|P_{1_{\mathcal{H}}}X_{\mathcal{H}}^T - P_{1_{\mathcal{H}}}X_{\mathcal{H}}^0\| \leq \sum_{t=1}^{T} d_t \leq \frac{\sqrt{\lambda}}{1-\sqrt{\alpha}}\sigma_0$$

Then, we use that $0 \leq \frac{1}{1-\sqrt{1-x}} \leq \frac{2}{x}$ for $x \geq 0$, with $x = \gamma(1-\delta)$.

$\square$

## C.4 Convergence of gossip NNA and ClippedGossip.

The previous $CG^+$ analysis can actually be performed exactly in the same way for the gossip version of nearest neighbors averaging, and to analyze ClippedGossip. Indeed, the aggregation rule and the impact of Byzantine nodes derives entirely from Lemma 6, and adapting this lemma to our gossip version of NNA and to ClippedGossip allows to derives convergence results. These results are close to the convergence results of $CG^+$, the only difference being the loss of a multiplicative factor 2 in the influence of Byzantine nodes (i.e everything happens as if the number of Byzantine was $2b$ instead of $b$).

### C.4.1 Case of gossip-NNA

A gossip version of NNA can be considered: instead of clipping the $2b$ farthest neighbors, each honest node $i$ removes the $b$ farthest neighbors. This leads to the following result:

**Corollary 4.** *Let $b$ and $\mu_{\min}$ be such that $8b \leq \mu_{\min}$, and let $\mathcal{G} \in \Gamma_{\mu_{\min},b}$. Then, assuming $\eta \leq \mu_{\max}(\mathcal{G}_{\mathcal{H}})^{-1}$, the output $\boldsymbol{y} = \text{NNA}(\boldsymbol{x})$ (obtained by one step of NNA on $\mathcal{G}$ from $\boldsymbol{x}$) verifies:*

$$\frac{1}{|\mathcal{H}|} \sum_{i \in \mathcal{H}} \|\boldsymbol{x}_i^{t+1} - \overline{\boldsymbol{x}}_{\mathcal{H}}^t\|^2 \leq (1 - \eta(\mu_{\min} - 8b)) \frac{1}{|\mathcal{H}|} \sum_{i \in \mathcal{H}} \|\boldsymbol{x}_i^t - \overline{\boldsymbol{x}}_{\mathcal{H}}^t\|^2 \tag{16}$$

$$\|\overline{\boldsymbol{x}}_{\mathcal{H}}^{t+1} - \overline{\boldsymbol{x}}_{\mathcal{H}}^t\|^2 \leq \eta \frac{8b}{|\mathcal{H}|} \sum_{i \in \mathcal{H}} \|\boldsymbol{x}_i^t - \overline{\boldsymbol{x}}_{\mathcal{H}}^t\|^2 \tag{17}$$

To prove Corollary 4, we only need to change Lemma 6 to adapt if for controlling the error due to NNA. The proof hinges on the following lemma, which we state and prove first, then we will prove the equivalent of Lemma 6 in the case of NNA.

**Lemma 7** (NNA: Control of the error). *The error due to removing honest nodes and due to Byzantine nodes is controlled by the heterogeneity as measured by the gossip matrix.*

$$\|\boldsymbol{E}^t\|_2^2 \leq 8b\|\boldsymbol{X}_{\mathcal{H}}^t\|_{\boldsymbol{W}_{\mathcal{H}}}^2 = 4b \sum_{i \in \mathcal{H}, j \in n_{\mathcal{H}}(i)} w_{ij} \|\boldsymbol{x}_i^t - \boldsymbol{x}_j^t\|^2$$

*Proof.* In this setting the error term writes

$$\forall i \in \mathcal{H}, \quad [\boldsymbol{E}^t]_i := \sum_{j \in n_{\mathcal{H}}(i)} w_{ij} \left(\boldsymbol{x}_i^t - \boldsymbol{x}_j^t - (\boldsymbol{x}_i^t - \boldsymbol{x}_j^t)\mathbf{1}_{j \text{ not removed}}\right) + \sum_{j \in n_{\mathcal{B}}(i)} w_{ij} \text{Clip}(\boldsymbol{x}_j^t - \boldsymbol{x}_i^t; \tau_i^t))$$

Applying the triangle inequality, we get

$$\|\boldsymbol{E}^t\|^2 = \sum_{i \in \mathcal{H}} \left\| \sum_{j \in n_{\mathcal{H}}(i)} w_{ij}(\boldsymbol{x}_i^t - \boldsymbol{x}_j^t)\mathbf{1}_{j \text{ removed}} + \sum_{j \in n_{\mathcal{B}}(i)} w_{ij}(\boldsymbol{x}_i^t - \boldsymbol{x}_j^t)\mathbf{1}_{j \text{ not removed}} \right\|_2^2$$

$$\leq \sum_{i \in \mathcal{H}} \left( \sum_{j \in n_{\mathcal{H}}(i)} w_{ij}\|\boldsymbol{x}_i^t - \boldsymbol{x}_j^t\|\mathbf{1}_{j \text{ removed}} + \sum_{j \in n_{\mathcal{B}}(i)} w_{ij}\|\boldsymbol{x}_i^t - \boldsymbol{x}_j^t\|_2\mathbf{1}_{j \text{ not removed}} \right)^2$$

By considering that node $i \in \mathcal{H}$ removes the largest values within $\{\|\boldsymbol{x}_i^t - \boldsymbol{x}_j^t\|; j \in n(i)\}$ until the total weight removed is strictly greater than $b$, we can consider for any choice of the Byzantine nodes, that a mass $k$ of honest neighbors is removed, and a mass $b - k$ of Byzantine neighbors are removed. As such, we can use the technical Lemma 8, where we denoted $a_1 \geq \ldots \geq a_{|n_{\mathcal{H}}(i)|}$ the sorted values within $\{\|\boldsymbol{x}_i^t - \boldsymbol{x}_j^t\|; j \in n_{\mathcal{H}}(i)\}$, and $\sum_{\text{weight} \leq b} a_i = \sum_{i=1}^{k_b} a_i$ where $k_b$ is the smallest index such that for all $j$, $\sum_{i=1}^{k_b} W_{ji} \geq b$.

$$\|\boldsymbol{E}^t\|^2 \leq \sum_{i \in \mathcal{H}} \left( 2 \sum_{\substack{j \in n_{\mathcal{H}}(i) \\ \text{weight} \leq b}} w_{ij} \|\boldsymbol{x}_i^t - \boldsymbol{x}_j^t\| \right)^2.$$

Which, using Cauchy-Schwarz inequality, yields

$$\|\boldsymbol{E}^t\|^2 \leq \sum_{i \in \mathcal{H}} 4b \sum_{j \in n_{\mathcal{H}}(i)} w_{ij}\|\boldsymbol{x}_i^t - \boldsymbol{x}_j^t\|^2.$$

The final result derives from the fact that $2\|\boldsymbol{X}_{\mathcal{H}}^t\|_{\boldsymbol{W}_{\mathcal{H}}}^2 = \sum_{i \in \mathcal{H}, j \in n_{\mathcal{H}}(i)} w_{ij}\|\boldsymbol{x}_i^t - \boldsymbol{x}_j^t\|^2$.

**Lemma 8** (NNA: Technical lemma for controlling $\boldsymbol{E}^t$). *Let $a_1 \geq \ldots \geq a_n \geq 0$, and $\mathrm{err}(k) := \sum_{weight \leq w} W_{ji}a_i + wa_{k_w}$, where $w \leq b$. Then*

$$\mathrm{err}(k) \leq 2 \sum_{weight \leq b} W_{ji}a_i.$$

*Proof.* We write

$$\mathrm{err}(k) = 2 \sum_{weight \leq w} W_{ji}a_i - \sum_{weight \leq w} W_{ji}(a_i - a_{k_w}) \leq 2 \sum_{i=1}^{b} W_{ji}a_i$$

Where we used that $a_i - a_{k_w} \geq 0$ for $i$ such that weight $\leq k$. $\qquad\square$

$\square$

### C.4.2 CASE OF CLIPPEDGOSSIP

As for sparse-NNA, ClippedGossip with his oracle clipping rule can be analyzed using the same approach. Then only difference being how the Lemma 6 is proven. As for NNA, the convergence result is only suboptimal to the one of $\mathrm{CG}^+$ by a factor 2, note however that ClippedGossip assumes to consider as clipping threshold for node $i \in \mathcal{H}$ at step $t$

$$\tau_i^t := \sqrt{\frac{1}{b} \sum_{j \in n_{\mathcal{H}}(i)} \|\boldsymbol{x}_i^t - \boldsymbol{x}_j^t\|^2}.$$

Which essentially consists in computing some average of the norms of *the honest parameters* differences, which is not possible in practice.

**Corollary 5.** *Let $b$ and $\mu_{\min}$ be such that $8b \leq \mu_{\min}$, and let $\mathcal{G} \in \Gamma_{\mu_{\min}, b}$. Then, assuming $\eta \leq \mu_{\max}(\mathcal{G}_{\mathcal{H}})^{-1}$, the output $\boldsymbol{y} = \mathrm{ClippedGossip}(\boldsymbol{x})$ (obtained by one step of ClippedGossip on $\mathcal{G}$ from $\boldsymbol{x}$) verifies:*

$$\frac{1}{|\mathcal{H}|} \sum_{i \in \mathcal{H}} \|\boldsymbol{x}_i^{t+1} - \overline{\boldsymbol{x}}_{\mathcal{H}}^t\|^2 \leq (1 - \eta(\mu_{\min} - 8b)) \frac{1}{|\mathcal{H}|} \sum_{i \in \mathcal{H}} \|\boldsymbol{x}_i^t - \overline{\boldsymbol{x}}_{\mathcal{H}}^t\|^2 \tag{18}$$

$$\|\overline{\boldsymbol{x}}_{\mathcal{H}}^{t+1} - \overline{\boldsymbol{x}}_{\mathcal{H}}^t\|^2 \leq \eta \frac{8b}{|\mathcal{H}|} \sum_{i \in \mathcal{H}} \|\boldsymbol{x}_i^t - \overline{\boldsymbol{x}}_{\mathcal{H}}^t\|^2 \tag{19}$$

**Lemma 9** (ClippedGossip: Control of the error). *The error due to the oracle rule of clipping the nodes and due to Byzantine nodes in ClippedGossip is controlled by the heterogeneity as measured by the gossip matrix:*

$$\|\boldsymbol{E}^t\|_2^2 \leq 8b\|\boldsymbol{X}_{\mathcal{H}}^t\|_{\boldsymbol{W}_{\mathcal{H}}}^2 = 4b \sum_{i \in \mathcal{H}, j \in n_{\mathcal{H}}(i)} \|\boldsymbol{x}_i^t - \boldsymbol{x}_j^t\|^2.$$

*Proof.* In this setting the error term writes

$$\forall i \in \mathcal{H}, \quad [\boldsymbol{E}^t]_i := \sum_{j \in n_{\mathcal{H}}(i)} \left(\boldsymbol{x}_i^t - \boldsymbol{x}_j^t - \mathrm{Clip}(\boldsymbol{x}_i^t - \boldsymbol{x}_j^t; \tau_i^t)\right) + \sum_{j \in n_{\mathcal{B}}(i)} \mathrm{Clip}(\boldsymbol{x}_j^t - \boldsymbol{x}_i^t; \tau_i^t))$$

By applying the triangle inequality, and by denoting $(a)_+ = \max(a, 0)$, we get

$$\|\boldsymbol{E}^t\|^2 = \sum_{i \in \mathcal{H}} \left\| \sum_{j \in n_{\mathcal{H}}(i)} \boldsymbol{x}_i^t - \boldsymbol{x}_j^t - \text{Clip}(\boldsymbol{x}_i^t - \boldsymbol{x}_j^t; \tau_i^t) + \sum_{j \in n_{\mathcal{B}}(i)} \text{Clip}(\boldsymbol{x}_i^t - \boldsymbol{x}_j^t; \tau_i^t) \right\|_2^2$$

$$\leq \sum_{i \in \mathcal{H}} \left( \sum_{j \in n_{\mathcal{H}}(i)} \|\boldsymbol{x}_i^t - \boldsymbol{x}_j^t - \text{Clip}(\boldsymbol{x}_i^t - \boldsymbol{x}_j^t; \tau_i^t)\|_2 + \sum_{j \in n_{\mathcal{B}}(i)} \|\text{Clip}(\boldsymbol{x}_i^t - \boldsymbol{x}_j^t; \tau_i^t)\|_2 \right)^2$$

$$\leq \sum_{i \in \mathcal{H}} \left( \sum_{j \in n_{\mathcal{H}}(i)} \left( \|\boldsymbol{x}_i^t - \boldsymbol{x}_j^t\|_2 - \tau_i^t \right)_+ + b\tau_i^t \right)^2.$$

Considering that

$$\left( \|\boldsymbol{x}_i^t - \boldsymbol{x}_j^t\| - \tau_i^t \right)_+ = \tau_i^t \left( \frac{\|\boldsymbol{x}_i^t - \boldsymbol{x}_j^t\|}{\tau_i^t} - 1 \right)_+$$

$$\leq \tau_i^t \left( \left( \frac{\|\boldsymbol{x}_i^t - \boldsymbol{x}_j^t\|}{\tau_i^t} \right)^2 - 1 \right)_+$$

$$\leq \tau_i^t \left( \frac{\|\boldsymbol{x}_i^t - \boldsymbol{x}_j^t\|}{\tau_i^t} \right)^2 = \frac{\|\boldsymbol{x}_i^t - \boldsymbol{x}_j^t\|^2}{\tau_i^t},$$

the error becomes:

$$\|\boldsymbol{E}^t\|^2 \leq \sum_{i \in \mathcal{H}} \left( \sum_{j \in n_{\mathcal{H}}(i)} \frac{\|\boldsymbol{x}_i^t - \boldsymbol{x}_j^t\|^2}{\tau_i^t} + b\tau_i^t \right)^2.$$

Minimizing each squared term within the sum leads to consider as clipping threshold

$$\tau_i^t := \sqrt{\frac{1}{b} \sum_{j \in n_{\mathcal{H}}(i)} \|\boldsymbol{x}_i^t - \boldsymbol{x}_j^t\|^2}.$$

Which gives the upper bound

$$\|\boldsymbol{E}^t\|^2 \leq \sum_{i \in \mathcal{H}} \left( \sum_{j \in n_{\mathcal{H}}(i)} \frac{\|\boldsymbol{x}_i^t - \boldsymbol{x}_j^t\|^2}{\tau_i^t} + b\tau_i^t \right)^2$$

$$= \sum_{i \in \mathcal{H}} \left( 2\sqrt{b \sum_{j \in n_{\mathcal{H}}(i)} \|\boldsymbol{x}_i^t - \boldsymbol{x}_j^t\|^2.} \right)^2$$

$$= 4b \sum_{i \in \mathcal{H},\, j \in n_{\mathcal{H}}(i)} \|\boldsymbol{x}_i^t - \boldsymbol{x}_j^t\|^2.$$

This corresponds to the desired result : $\|\boldsymbol{E}^t\|^2 \leq 8b\|\boldsymbol{X}_{\mathcal{H}}\|_{\boldsymbol{W}_{\mathcal{H}}}^2$ □

**Remark 1.** *A key point here is that this oracle clipping threshold corresponds to the **unique minimizer** within each squared term of the sum. Hence, considering for instance the adaptive practical clipping rule of (He et al., 2022) leads to a **larger upper bound on the error**.*

C.5 PROOFS FOR $D - SGD$

*Proof of Corollary 2.* This proof hinges on the fact that the proof of Farhadkhani et al. (2023, Theorem 1) does not actually require that communication is performed using NNA, but simply that

the aggregation procedure respects $(\alpha, \lambda)$-reduction, which they prove in their Lemma 2. Then, all subsequent results invoke this Lemma instead of the specific aggregation procedure. $CG^+$ also satisfies $(\alpha, \lambda)$-reduction, as we prove in Theorem 2. We can then use the bounds on the errors out of the box.

Then, as $T$ grows, and ignoring constant factors, only the first and last terms in their Theorem 3 remain, leading to:

$$\sum_{t=1}^{T} \mathbb{E}\left[\left\|\nabla f_{\mathcal{H}}(\boldsymbol{x}_i^t)\right\|^2\right] = \mathcal{O}\left(\frac{L\sigma}{\sqrt{T}}(1+C) + \zeta^2 C\right), \tag{20}$$

where $C = c_1 + \lambda + \lambda c_1$, with $c_1 = \alpha(1+\alpha)/(1-\alpha)^2$. Note that we give $\mathcal{O}()$ versions of the Theorems for simplicity, but Farhadkhani et al. (2023, Theorem 1) allows to derive precise upper bounds for any $T \geq 1$.

**One-step derivations.** The one-step result is obtained by taking the values of $\alpha = 1 - \gamma(1-\delta)$ and $\lambda = \gamma\delta$, and considering $\gamma < 1$ (otherwise, the guarantees are essentially the same as in Farhadkhani et al. (2023)). More specifically:

$$c_1 = \frac{(1-\gamma(1-\delta))(2-\gamma(1-\delta))}{\gamma^2(1-\delta)^2} = O\left(\frac{1}{\gamma^2(1-\delta)^2}\right). \tag{21}$$

Meanwhile, $\lambda = \gamma\delta \leq 1$, so that $C = O(c_1)$, leading to the result.

**Multi-step derivations.** In the previous case, we see that $C$ is dominated by the $c_1$ term since $c_1 >> \lambda$. In particular, the guarantees would increase if we were able to trade-off some $\alpha$ for some $\lambda$, which is possible by using multiple communications steps. This is what we do, and take enough steps that $c_1 << \lambda$ (i.e., $\alpha \approx 0$), so that $C \approx \lambda$. Following Corollary 1, this requires $\tilde{O}(\gamma^{-1}(1-\delta)^{-1})$ steps, where logarithmic factors are hidden in the $\tilde{O}$ notation. We then plug the multi-step $\lambda$ value from Corollary 1 to obtain the result.

$\square$

## D  LINKS BETWEEN DEFINITIONS OF GOSSIP MATRICES

In this work, we defined gossip algorithms based on weighted Laplacian of graphs. This is in line of several work such as Scaman et al. (2017); Kovalev et al. (2020). Yet some other works rely on symmetric bistochastic gossip matrices, such as Koloskova et al. (2019); He et al. (2022). Both definitions have deep connections, without always being strickly equivalent. We state here the following two definitions, then we show how to go from one to the other.

**Definition 3** (Bistochastic gossip matrix). *A matrix $\boldsymbol{B}$ is said to be a bistochastic gossip matrix of an undirected connected graph $\mathcal{G}$ if*

- *$\boldsymbol{B} \in [0,1]^{m \times m}$ is bistochastic symmetric, ie $\boldsymbol{B}^T = \boldsymbol{B}$ and $\boldsymbol{B}\mathbf{1} = \mathbf{1}$.*

- *$\boldsymbol{B}_{ij} \neq 0$ if, and only if, $i \neq j$ or $i \in n(j)$*

**Definition 4** (Non-negative gossip matrix). *A matrix $\boldsymbol{W}$ is a non-negative gossip matrix of an undirected connected graph $\mathcal{G}$ if*

- *$\boldsymbol{W} \in \mathbb{R}^{m \times m}$ is symmetric non-negative.*

- *The kernel of $\boldsymbol{W}$ is restricted to the span of the constant vector: $\ker \boldsymbol{W} = \mathrm{span}(\mathbf{1})$.*

- *$\boldsymbol{W}_{ij} \neq 0$ if, and only if, $i \neq j$ or $i \in n(j)$.*

The following proposition provide a way to go from one definition to the other.

**Proposition 1.**

- *Let $\boldsymbol{B}$ a bistochastic gossip matrix, then $\boldsymbol{I} - \boldsymbol{B}$ is a non-negative gossip matrix.*

- *Let $\boldsymbol{W}$ a non-negative gossip matrix with non-positive non-diagonal coefficients, then under $\eta \leq \mu_{\max}(\boldsymbol{W})$, the matrix $\boldsymbol{I} - \eta\boldsymbol{W}$ is a bistochastic gossip matrix.*

