# OpenReview forum: "Achieving Optimal Breakdown for Byzantine-Robust Gossip"
_ICLR.cc/2025/Conference — Submitted to ICLR 2025_

### Official Review · Reviewer_mDqu · 2024-10-18

**Soundness:** 1
**Presentation:** 2
**Contribution:** 2
**Rating:** 3
**Confidence:** 4

**Summary:**

This paper analyzes the breakdown point of robust algorithms in decentralized optimization and proposes the CG Plus method, which achieves the optimal breakdown point. Convergence guarantees and experimental results are provided to support the proposed method.

**Strengths:**

1. The analysis of the breakdown point for robust algorithms in decentralized optimization is novel and has not been previously explored.


2. The proposed CG Plus is well-motivated, addressing the impractical clipping threshold issue in ClippedGossip.

**Weaknesses:**

1. The experimental evaluation is insufficient. The authors only conducted experiments on the MNIST dataset, and additional results from other datasets are recommended to strengthen the conclusions.

2. The experimental results do not clearly demonstrate the advantage of CG Plus. In fact, ClippedGossip seems to outperforms the proposed CG Plus overall. The authors' claim that "ClippedGossip might fail against other attacks" is unconvincing. It is recommended that further results be provided to demonstrate CG Plus's advantage over ClippedGossip; otherwise, the development of CG Plus seems unjustified.

**Questions:**

1. The authors utilize the Laplacian matrix of the graph as the gossip matrix. However, in decentralized optimization, the gossip matrix is typically required to be doubly stochastic (He et al., 2022). Moreover, in Section 4.1, the weighted averaging step with $W_{ij}$ in ClippedGossip  (He et al., 2022) is replaced by a "communication step-size" $\eta$. The condition $\eta \leq 1/u_{max}$ seems unable to recover traditional averaging when there are no Byzantine agents. The authors should provide more explanation regarding $\eta$ and clarify why they chose not to use the conventional doubly stochastic gossip matrix.

2. The paper investigates the breakdown point in terms of the number of Byzantine agents. However, in traditional Byzantine-robust decentralized optimization, the focus often lies on the gossip weights assigned to Byzantine agents, rather than merely the number of Byzantine agents (He et al., 2022). For instance, if more than half of the neighbors are Byzantine, but the total gossip weight assigned to them is minimal, the algorithm can still remain robust. Does this perspective conflict with Theorem 1? It is recommended that the breakdown point analysis consider the gossip matrix in more detail.

---

> ### Author Response · Authors · 2024-11-20
> **Answer to Reviewer**
>
> We thank you for your review and questions.
>
> ## General comment on theory vs Experiments.
>
> We respectfully disagree that the fact that one of the methods we introduce is experimentally on par with competing methods constitutes a valid ground to reject our paper.
>
> Both weaknesses you underlined regard experiments, and disregard the theoretical aspect of our paper, which is our main contribution. Security in decentralized learning can only advance by being supported by theoretical guarantees, and will be built on shaky foundations otherwise, being exposed to the risk of new attacks beating existing robust rules.
>
> ## Detailed Answer
>
> In our paper, we focused on providing an analysis of robust gossip algorithms with tight theoretical guarantees, as it is still lacking in this field of research, rather than benchmarking the different approaches.
>
> For instance, out of the 2 previous papers we investigated, *no implemented algorithms had any convergence guarantees in the gossip decentralized setting* (i.e on sparse graphs). We are the first to provide both experimental evaluation and theoretical validation of implementable algorithms.
>
> As you point out, the original ClippedGossip corresponds either to a non-implementable algorithm with sub-optimal convergence guarantees, or to a competitive algorithm with no theoretical foundations.
>
> Still, following your concerns, we added experiments on CIFAR-10 as well as an extensive benchmark on MNIST dataset. In this latter benchmark, we plotted the resulting loss and accuracy under a varying number of Byzantine agents.
>
> Our new experiments show that CG+ clearly outperforms the gossip adaptation of NNA, in the sense that it is robust up to a larger amount of Byzantines. Consequently, CG+ seems to be the more robust of the aggregation rules supported by theory (as the implementable version of ClippedGossip is not).
> Furthermore, we show that CG+ has similar performances than ClippedGossip below the breakdown point. Interestingly, the line search approach to design the scale of attacks, as done in [1] fails to make CG+ break (while ClippedGossip does break), even after the breakdown point of our Theorem 1.
>
> We kindly ask you to reconsider your score to take into account that the main goal of our paper is to provide **theoretical foundations for robust gossip algorithms**, and that our experiments are mostly meant to illustrate and validate these results. Theory is very important for robust algorithms since experimental performance highly depends not only on the chosen dataset but also on the considered attacks, which can change rapidly.
>
> **Q1**: Our choice of instantiating all results with the graph’s Laplacian was motivated only by simplicity of the paper exposition. As you point out indeed, traditional local averaging can be recovered using the graph Laplacian only for $d$-regular graphs (by choosing as communication step size $\eta = 1/(d + 1)$).
>
>
> Generically, there are two standard ways of defining gossip matrices, one which is the choice taken for instance in [2] and [3] based on bistochastic matrices, and the other one which relies on non-negative symmetric matrices with kernel restricted to the constant vector, as done in [4,5].
>
> We believe that considering non-negative matrices as gossip matrices makes more sense for the robust gossip algorithms studied, as 1) they rely on computing differences between parameters, which is directly performed with the non-negative definition of gossip matrices and 2) the identified lower bound precisely corresponds to an eigenvalue of the non-negative gossip matrix.
>
>
> **Q2**: Exposing the paper with a number of Byzantine neighbors, instead of weights, was just for the sake of simplicity. Indeed, it is straightforward to extend both our Theorem 1 and Theorem 2 in the case of Laplacian matrices with generic weights on each edge of the graph. We implemented this generalization in our paper.
>
> For instance, performances of algorithms are not studied anymore on the class of graphs $\Gamma_{\mu_{\min}, :b}$, but on pair of graph and gossip matrix belonging to a new class $(G, W) \in \Gamma_{\mu_{min}, \: b}$, where $b$ is un upper bound on the weight associated with Byzantine neighbors of any honest node.
>
>
> Note that any symmetric bistochastic matrix $W$ can be turned into a graph Laplacian with non unitary weights, by doing  $L = Id - W$. Hence our generalization allows us to encapsulate the bistochastic definition of gossip matrices. We provided a note in the appendix to enlighten the links between bistochastic and non-negative symmetric matrices.
>
> We would be glad to provide any further clarification needed.

---

> > ### Author Response · Authors · 2024-11-20
> > **References**
> >
> > [1]: Sadegh Farhadkhani, Rachid Guerraoui, Nirupam Gupta, Leˆ-Nguyeˆn Hoang, Rafael Pinot, and John Stephan. Robust collaborative learning with linear gradient overhead. In International Conference on Machine Learning, pp. 9761–9813. PMLR, 2023.
> >
> >
> > [2]: Lie He, Sai Praneeth Karimireddy, and Martin Jaggi. Byzantine-robust decentralized learning via clippedgossip. arXiv preprint arXiv:2202.01545, 2022.
> >
> >
> > [3] Anastasia Koloskova, Sebastian Stich, and Martin Jaggi. Decentralized stochastic optimization and gossip algorithms with compressed communication. In International Conference on Machine Learning, pp. 3478–3487. PMLR, 2019.
> >
> >
> > [4] Kevin Scaman, Francis Bach, Se ́bastien Bubeck, Yin Tat Lee, and Laurent Massoulie ́. Optimal algorithms for smooth and strongly convex distributed optimization in networks. In international conference on machine learning, pp. 3027–3036. PMLR, 2017
> >
> > [5] Dmitry Kovalev, Adil Salim, and Peter Richta ́rik. Optimal and practical algorithms for smooth and strongly convex decentralized optimization. Advances in Neural Information Processing Systems, 33:18342–18352, 2020

---

> > > ### Comment · Reviewer_mDqu · 2024-11-25
> > >
> > > Thank you for the author’s response. I understand that as a theoretical paper, some experiments may be limited. However, even setting aside the experimental results, the theoretical contributions are not entirely satisfactory to me.
> > >
> > > 1. The authors do not provide the breakdown points for general robust decentralized learning algorithms (with a doubly stochastic mixing matrix). I could not find this information in either the revised manuscript or the response.
> > > 2. The authors stated: "No implemented algorithms had any convergence guarantees in the gossip decentralized setting (i.e., on sparse graphs). We are the first to provide both experimental evaluation and theoretical validation of implementable algorithms."
> > > This claim is overstated and not true. Numerous works have already provided theoretical guarantees and experimental results for Byzantine-robust decentralized learning, which the authors have overlooked. For instance:
> > >
> > > •	Zhaoxian Wu, Tianyi Chen, and Qing Ling. Byzantine-resilient decentralized stochastic optimization with robust aggregation rules. IEEE Transactions on Signal Processing, 2023.
> > >
> > > •	Cheng Fang, Zhixiong Yang, and Waheed U Bajwa. BRIDGE: Byzantine-resilient decentralized gradient descent. IEEE Transactions on Signal and Information Processing over Networks, 8:610–626, 2022.
> > >
> > > •	Yang C, Ghaderi J. Byzantine-Robust Decentralized Learning via Remove-then-Clip Aggregation. Proceedings of the AAAI Conference on Artificial Intelligence, 2024, 38(19): 21735-21743.
> > >
> > > If the authors wish to claim their convergence results are not optimal in terms of breakdown points, they should explicitly provide the breakdown points of these algorithms and compare them with the lower bound of breakdown points established for a general class of robust algorithms.
> > >
> > > If the authors address my concerns, I will reevaluate this submission.

---

> ### Author Response · Authors · 2024-11-27
> **Breakdown points using bistochastic matrices**
>
> Thank you for your precise and relevant response.
>
> # Question 1.
>
> As you point out, going beyond the Laplacian matrix and directly providing the optimal breakdown point for bistochastic matrix is of significant interest for many. Thus, we provide below the different breakdown points in our article using bistochastic matrices. We will add this discussion to our appendix.
>
> *Notations.* To do it, we use the following notations: We denote as $W$ a symmetric bistochastic gossip matrix, and $L$ a Laplacian matrix of the graph (e.g., $L = I - W$). We denote $w_B \ge \sum_{j \in n_B(i)}W_{ij} \in [0,1]$ the maximal total weight associated with Byzantines in the neighborhood of any node $i$. Eventually $\gamma(W_H) = 1 - \max(\mu_{H-1}(W_H), - \mu_1(W_H))$ denotes the spectral gap of the bistochastic honest gossip matrix $W_H$. Eigenvalues of matrices are ordered in an increasing manner: $\mu_1 \le \ldots \le \mu_H$.
>
> ## Upper bound
>
> > **Theorem 1.** *(w. bistochastic gossip matrix)* For any $\gamma \ge 0$, and any $w_B \in [0,1]$, if $w_B \ge 2\gamma$, there exist for any $H \ge 0$ a graph $G$ of $H$ honest nodes and an associated gossip matrix $W$, where $\gamma(W_H) = \gamma$ such that no algorithm is $\alpha$-robust on $G$.
>
> In other words, instead of the second-smallest eigenvalue of the Laplacian matrix, the breakdown point of robust gossip algorithms with bistochastic matrices is expressed using $\gamma(W_H)$, the spectral gap of the gossip matrix of the honest subgraph.
>
> Note that one goes from $W$ to $W_H$ in the following way: if $i \\neq j$, the index $(i,j)$ of $W_H$ is equal to $W_{ij}$, while on the diagonal $W_H$ is equal to $(W_{ii} + \sum_{j \in n_B(i)} W_{ij})_{i=1, \ldots, H}$.
>
> >**Proof**. The proof relies on exactly the same graph as the *Theorem 1*, on which no $\alpha$-robust algorithm is possible. The gossip matrix considered is $W = I - \eta L$, where $L$ is the (unitary weighted) laplacian matrix of the considered graph.
> >- On the one hand, the weight of Byzantines in the neighborhood of each honest node is equal to $w_B = \eta b$, where $b$ is the number of Byzantine neighbors of honest nodes in the graph.
> >- On the other hand, on the considered graph there is $\mu_{2}(L_H) = 2b$ according to proof in the paper. Furthermore its eigenvalues between $L_H$ and $W_H$ are linked as follows: $\eta \mu_{2}(L_H) = 1 - \mu_{H-1}(W_H)$. Note that under $\eta \le 1/\mu_{H}(L_H)$, we have $\mu_{1}(W_H)\ge 0$. Hence the spectral gap of $W_H$ is $\gamma = 1 - \mu_{H-1}(W_H) = \eta \mu_2(L_H)$, i.e $\gamma = 2b\eta$.
> >  Putting things together leads to a graph where $\gamma = 2w_B$, and on which there is no $\alpha$-robust algorithm. Note that choosing properly $\eta$ allows to enforce that $b$ is an integer. This concludes the proof.
>
> ## Breakdown point of each algorithm.
>
> In our article, we provide breakdown points of CG+, gossip NNA and ClippedGossip by relying on a Laplacian gossip matrix L. The breakdown points of our article are expressed as $c w_B \le \mu_{2}(L_H)$, where the constant $c$ depends on the considered algorithm.
>
> Assume that the Laplacian matrix of the graph derives from a bistochastic gossip matrix $W$ through $W = I - L$. Then the previous breakdown point writes $cw_B \le 1 - \mu_{H-1}(W_H)$. Remarking that $\gamma(W_H) \le 1 - \mu_{H-1}(W_H)$ allows us to state all our theorems using as the breakdown points assumption $c w_B \le \gamma(W_H)$, where the constant $c$ depends on CG+, NNA or ClippedGossip.
>
> Note that, interestingly, this upper bound of the spectral gap shows that the second-smallest eigenvalue of the graph Laplacian gives a breakdown point a bit more precise than the spectral gap.
>
>
> We use this formulation of the breakdown point to compare ourselves to the articles you mentioned in the rest of our answer.

---

> ### Author Response · Authors · 2024-11-27
> **Comparison with existing works**
>
> ## Question 2
>
> Thank you for pointing out these articles. We mainly made our statement to highlight the differences of our work with NNA and ClippedGossip, and went a bit farther than what we intended to do. We apologize about that. Indeed, we did refer to [Wu et al. 2023] and [Fang et al. 2022] in our paper. We highlight below the improvement we made over the pointed-out papers, which we will clarify in our work.
>
> Note that - as you point out - these articles provide both experimental and theoretical guarantees, yet our work either tackles a very different setting or improves by a significant factor in the breakdown point over the cited article.
>
> *Notations.* For simpler comparison, we provide results with notations in the bistochastic gossip matrix setting. Denote by $w_B \in [0,1]$ the maximal weight of Byzantines in the neighborhood of any honest node. In each article, we identify a *breakdown ratio* denoted $\delta$, such that the convergence result holds if and only if $\delta \le 1$. We recall that $\gamma$ is the spectral gap of the honest gossip matrix, $\zeta^2$ the heterogeneity of the loss functions, $\sigma^2$ the variance of the stochastic oracles, and $H$ the number of honest nodes in the graph.
>
> | Article | Algorithm | losses  assumption |Breakdown ratio | Asymptotic error |
> | -------- | -------- | -------- | -------- | -------- |
> | Wu et al 2023.  | IOS | smooth  heterogeneous | $\delta = \frac{8w_B\sqrt{H}}{\gamma}$     | $\mathcal{O}\left(\frac{\delta^2 }{\gamma}(\sigma^2 + \zeta^2)\right)$    |
> | Fang et al 2022  | BRIDGE-T | lipschitz smooth strongly convex i.i.d datas   |   Combinatorial assumption   | high probability result     |
> | Ghaderi et al 2024 | ClippedGossip w. adaptive rule o NNA | smooth heterogeneous  | $\delta=\mathcal{O}(\frac{w_B}{\gamma^2})$     | $\mathcal{O}(\delta \zeta^2)$     |
> | Ours  | CG+    | smooth heterogeneous | $\delta=\frac{4w_B}{\gamma}$     | $\mathcal{O}(\frac{\delta}{\gamma} \zeta^2)$ |
>
> We now first compare each of these results, then we explain how to translate the result of their article into our notations.

---

> ### Author Response · Authors · 2024-11-27
> **Comparison**
>
> 1. [Wu et al. 2023].
>
>   - **Breakdown point.** Their breakdown ratio is suboptimal by a significant factor of $\sqrt{H}$; hence, when the number of honest nodes in the graph increases, it becomes harder to have a positive breakdown ratio, independently of the graph connectivity. For instance, in a fully connected network they can only handle a proportion of $\mathcal{O}(1/\sqrt{H})$ Byzantine nodes, when the optimal proportion is $1/3$.
>
>   - **Asymptotic error.** They have a square factor on $\delta$ with respect to our result. However, considering that their $\delta$ is significantly larger than ours for large graphs, there is only a gain when the size of the graph is fixed, or when the proportion of Byzantines vanishes to 0. Furthermore they rely on a stronger definition of $\zeta^2$, which can be strictly  greater than ours up to a factor $H$.
>
> Note that IOS is computationally more expensive than CG+, NNA and ClippedGossip. Indeed, for any honest node $i$, the cost of IOS is $\mathcal{O}(b|n(i)|d)$, while the cost of the others is $\mathcal{O}(|n(i)|d)$.
>
> We believe this article is close to our work, and we will add this discussion as well as experiments on IOS in the camera-ready version of the paper.
>
>
> 2. [Fang et al. 2022].
>
> Their article investigate a significantly different setting to ours:
>
> - It is assumed that data are sampled i.i.d. among all nodes, which essentially means that there is **no heterogeneity** among all nodes (other than due to finite samples). This is a very significant difference with respect to our setting, as it is less important to communicate.
>
> - They assume *strongly convex Lipschitz smooth functions* (or rely on local strict convexity), while we consider more generic, *non-necessarily convex smooth* functions.
>
> - They use an assumption on the connectivity of the graph, which does not directly translate into gossip matrix quantities. It is unclear whether this assumption could be useful in the case of the heterogeneous loss functions. For instance, [Farhadkani et al. 2023] and [He et al. 2022] reported bad experimental performances of BRIDGE in the heterogeneous setting.
>
> Considering the gap in the problem investigated, and the previous experimental comparison done by other papers, we believe that further experiments using BRIDGE do not bring additional knowledge to the community.
>
> 3. [Ghaderi et al. 2024].
>
> This article essentially provides theoretical foundations and a fix to the practical adaptive rule of ClippedGossip. They achieve it by composing the adaptive clipping threshold of [He et al.] with NNA.
>
> - **Breakdown point.** Their convergence result requires that $w_b \in \mathcal{O}(\gamma^2)$, which is suboptimal, as we show in our paper.
>
> - **Convergence result.** Their convergence result is the same as [He et al.], hence they have similar performance guarantees as us, considering that our breakdown ratio $\delta$ differ by a factor $\gamma$.
>
> This paper is quite recent (2024), thus we did not know about it. Unfortunately, we could not find the supplementary materials of the paper online (including the proofs), and we would be highly interested if you knew how to find them. Not least because there was an error in [He et al. 2022] proof, and we'd like to know how they went about fixing it.

---

> > ### Author Response · Authors · 2024-11-27
> > **Translation of the cited work to our notations**
> >
> > First, we omit polynomial dependences in $\mathcal{O}(1/(1-\delta))$ in the asymptotic error, as they can always be removed by loosing a constant factor in the definition of $\delta$.
> >
> > 1. [Wu et al. 2023].
> >
> > The author relies on two properties of the considered aggregation rules for providing convergence results: The *Robust Contractive Aggregation* (RCA) property, which identifies the breakdown point in their analysis, and the *Robust Doubly Stochastic Aggregation* (RDSA) property, which states that the communication is "close" to a gossip communication with a doubly-stochastic matrix.
> >
> > - **Breakdown Point.** The RCA property requires (in their notation) that $\lambda/8\sqrt{N} > \rho$. Where in $\lambda$ corresponds to the spectral gap ($\gamma$ in our notations), $\rho$ corresponds to $w_B$ in the case of IOS, and $N=H$ the number of honest nodes in the graph. Hence, their breakdown ratio writes $\delta=\frac{8 w_B \sqrt{H}}{\gamma}$.
> >
> > - **Convergence Result.** We consider their final result on the asymptotic error (Corollary 1). They denote by $\delta_{in}$ (resp. $\delta_{out}$) the variance of the stochastic gradients $\sigma$ (resp. the heterogeneity of the loss functions $\zeta$). Furthermore, we can link their $\omega$ with our quantities as $\omega = \gamma(1-\delta)$ (using their specific breakdown ratio $\delta$). They rely on a third quantity, $\Delta$, which, considering the previous decomposition of $\omega$, is equal to $\Delta=\frac{1 - \gamma(1-\delta)}{\gamma^3(1-\delta)^3}$. Considering these translations of notation, their asymptotic error is upper bounded by
> > $$
> > error = \mathcal{O}\left(w_B^2 H \frac{1 - \gamma(1-\delta)}{\gamma^3(1-\delta)^3}(\sigma^2 + \zeta^2)\right) \in \mathcal{O}\left(\frac{\delta^2}{\gamma(1-\delta)^3}(\sigma^2 + \zeta^2)\right)
> > $$
> >
> >
> > 2. [Ghaderi et al. 2024].
> >
> > The only difference in notation is that they denote the maximal weight associated with Byzantines in the neighborhood of an honest node as $\delta_{\max}$, which we denote here as $w_B$.

---

> > > ### Comment · Reviewer_mDqu · 2024-11-29
> > >
> > > Thank you for the authors’ efforts in providing a response. I sincerely appreciate it. Below, I have outlined several comments for consideration:
> > >
> > > 1.	Tightness of Breakdown Points: In Theorem 1, the authors establish that $W_B \geq 2r$ represents the breakdown point. However, it appears that the breakdown point for CG-plus is $4W_B \geq r$, which does not seem optimal. This raises two questions:
> > >
> > > (a) Is the breakdown point in Theorem 1 tight?
> > >
> > > (b) Why is the breakdown point for CG-plus considered optimal in the context of the previous Laplacian matrix setting but not under the doubly stochastic setting? More detailed discussions on these aspects should be added.
> > >
> > > 2. Broader Discussions:
> > > Several works on Byzantine-robust decentralized learning suggest that the fraction of Byzantine agents cannot exceed 1/3. The authors should discuss this relationship with Theorem 1 and provide additional examples in special topologies (e.g., fully connected topology) to better illustrate the breakdown point.
> > >
> > > 3.	Breakdown Point of Clipped Gossip:
> > > The breakdown point of Clipped Gossip should be included in the response to Question 2, as CG-plus builds upon it.
> > >
> > > 4.	Relationship Between $\rho$ and $W_B$:
> > > It is unclear how $\rho$ in [Wu et al., 2023] corresponds to $W_B$. To my understanding, $\rho$ characterizes the RCA property of the aggregator, while $W_B$ represents the Byzantine weights. This connection requires further clarification.
> > >
> > >
> > > 5.	Highlighted Revisions:
> > > It is recommended that the revised sections in the submission be explicitly highlighted, as I could not locate the changes or identify useful updates.
> > >
> > > Based on the above points, I believe substantial work is needed. Therefore, I have decided to maintain my score at the current stage.

---

> ### Author Response · Authors · 2024-11-29
>
> We sincerely thank the reviewer for their prompt response and his willingness to engage in improving the paper.
>
> ## 1 - Tightness of the Breakdown points.
> The constant factors of breakdown points in the case of Laplacian matrix and the case of Bistochastic matrix **are exactly the same**, we appologize if our previous response was not clear on this point.
>
> We fear that the reviewer may have overlooked the global response, in which we point out a loss of a multiplicative factor 2 in the breakdown point of CG+ (Lemma 7). We show (in the revised version of the article) that CG+ can be exactly optimal when using an 'oracle' clipping threshold (oracle CG+), but using a non-oracle threshold (practical CG+) leads to this suboptimal multiplicative factor 2. By default, we referred in our answers to the practical version of CG+.
>
>
> Note that being suboptimal by a factor 2 does not change the fact that we improved on the previous results by **significant non-constant factors**. Note as well that we changed the title of the article and do not claim exact optimality anymore, and focus on the unified analysis aspect.
>
> **(a)** We believe that Theorem 1 is tight, as, in the case of fully connected graphs, Theorem 1 boils down to claiming that the maximal breakdown point is 1/3, which is known to be optimal.
>
> **(b)** As pointed out previously, constant factors are the same for the Laplacian matrix and bistochastic matrices point of view. Hence, on the light of Theorem 1 both approaches are just as (near-)optimal.
>
>
> ## 2 - Broader Discussion
> Our Theorem 1 strictly includes the case of a fully connected graph: The fully connected graph corresponds to the graph with spectral gap $\gamma(W_H)=1$. As such, the requirement of $w_B \le 2\gamma$ boils down to having $2$ times more honest nodes than Byzantine nodes in the graph, i.e., at most $1/3$ of Byzantines. We do give this result at line 209 of our paper.
>
> ## 3 - Breakdown of ClippedGossip
> We would like to make sure we correctly understand your concern: the Breakdown point of ClippedGossip is given in the General Response above (see the table, that is now Table 1 of the paper). In the response to your question 2, our answer was focused on the references that you had asked us to compare to (which helped improve the paper and will be included too in the final version).
>
>
>
> ## 4 - Relationship between $\rho$ and $w_B$.
> As the reviewer points out, Corollary 1 in [Wu et al. 2023] relies on a generic $\rho$ from the RCA property of aggregators. Nonetheless, [Wu et al. 2023] provide in their section VI.B an upper bound of $\rho$ in the case of their IOS rule. Which, using their notation, is the following:
>
> $$
> \rho \le \max_{n \in \mathcal{N}} \frac{15 \mathcal{W}_n'(\mathcal{U}_n^{max})}{1 - 3 \mathcal{W}_n'(\mathcal{U}_n^{max})}
> $$
>
> Here $\mathcal{W}_n'(\mathcal{U}_n^{max})$ denotes the maximal weight of $q_n$ neighbors of the node $i$ ($q_n$ is the number of Byzantines in the neighborhood of $i$). As such we always have that $\mathcal{W}_n'(\mathcal{U}_n^{max}) \ge w_B$. Therefore,taking
> $$
> \rho' = \frac{15w_B}{1 - 3w_B} \in \mathcal{O}(w_B)
> $$
> instead of the actual $\rho$ is to their advantage (i.e., gives an optimistic version of their result). For instance $\rho = \rho^\prime$ when all edges are equally weighted (but again, not in general).
>
> For simpler comparison, we used $\rho = w_B$ in the previous answer, thus underestimating their $\delta$ by a factor $15$, while neglecting the gap between $\mathcal{W}_n'(\mathcal{U}_n^{max})$ and $w_B$. This gap is particularly significant when edge weights vary substantially.
>
>
>
> ## 5 Highlighted revisions
>
> We sincerely apologize for the lack of highlighted changes in the revision.
>
> Normally, the  *Revision Comparison* option of OpenReview enables to do that. Another solution is to download the paper version before and after the rebuttal and to compare it using, for instance, diffchecker. If the reviewer wishes, we can also provide a detailed list of the changes made during the rebuttal.
>
> We provide below a clarification on the revisions planned and the ones already implemented.
>
> Note that due to time constraints, we only implemented in the current version available on OpenReview the revisions made during the first round of answers to reviewers, and in particular we did not have time to include the comparison with related work. For absolute completeness, we provide hereafter the planned revisions that summarize some aspects of the discussion above.

---

> > ### Author Response · Authors · 2024-11-29
> >
> > ### Planned Revisions
> >
> >
> > - Expand the appendix on the links between bistochastic gossip matrices and Laplacian matrices by adding clear breakdown points and convergence results for both Theorem 1 and rules such as NNA, ClippedGossip and CG+.
> > - Add a comparison between the breakdown point of IOS and CG+
> > - Improve the literature review by comparing our results to [Wu et al. 2023], [Fang et al. 2022] and [Ghaderi et al. 2024].
> > - Implement IOS aggregation rule and compare it experimentally to the other algorithms (NNA, CG+, ClippedGossip).
> >
> > Note that all points but the last (experimental) one are essentially included in the current responses to reviewers.
> >
> >
> > ----
> > Once again, we appreciate your valuable feedback, which has greatly contributed to strengthening the paper. We hope that these precision addresses your concern, and we welcome any additional questions you may have.

---

> > > ### Comment · Reviewer_mDqu · 2024-12-01
> > >
> > > Thank you for the authors’ response.
> > >
> > > If I understand correctly, according to Table I in the submission, CG+ improves the breakdown point by a factor of 2 compared with clipped gossip, while still remaining a factor of 2 away from the optimal breakdown point.  If the optimal breakdown point is 1/3, this implies that the breakdown point of CG+ is 1/6. When the number of agents is large, this gap becomes significant. I believe that such a factor of 2 cannot be overlooked as a mere constant, unlike typical constants in convergence results.
> > >
> > > Furthermore, I notice that CG+ with oracle achieves the optimal breakdown point, but it is not practical for real-world applications. Is it possible to design a practical algorithm that achieves the optimal breakdown point? Can the authors provide additional discussions on this topic?

---

> > > > ### Author Response · Authors · 2024-12-02
> > > >
> > > > Thank you for your comment.
> > > >
> > > > 1. You understand perfectly well, and we agree that a constant factor of 2 for the breakdown point is significant. This precise point is one of the main motivations for our article, as previous approaches had much lower breakdown points, making their algorithms difficult to apply in practice. Please note that the breakdown point of ClippedGossip is also one of our contribution, as we improve w.r.t [He et al. 2022] by specifying the constant factor and by removing a factor $1/\gamma$, which goes to infinity when the graph is less connected.
> > > >
> > > > 2. We do not exactly know how to achieve the optimal breakdown with a non-oracle rule. We believe that this is beyond the scope of our paper, and we let it for future work.
> > > >
> > > > We thank you again for your time and your consideration.

---

### Official Review · Reviewer_jJqZ · 2024-10-23

**Soundness:** 2
**Presentation:** 3
**Contribution:** 2
**Rating:** 6
**Confidence:** 4

**Summary:**

This paper analyzes the optimal breakdown point of robust aggregators in the Byzantine-robust decentralized average consensus problem and proves that the breakdown point of the proposed CG$^+$ method almost aligns with the optimal values. To further validate the effectiveness of the CG$^+$ method in general Byzantine-robust decentralized stochastic problems, this paper examines its theoretical convergence and demonstrates its practical performance compared to existing methods in experiments.

**Strengths:**

The analysis of the optimal breakdown point of robust aggregators is both novel and significant in the field of Byzantine-robust decentralized learning.

**Weaknesses:**

1. The lower bound of the breakdown point indicated in Theorem 1 appears to be not tight, as the breakdown point of CG$^+$ presented in Theorem 2 ($b < \frac{\mu_{\min}}{2} - 1$) does not align with the lower bound ($b \geq \frac{\mu_{\min}}{2}$). This leaves a gap when $b \in$ { $\lceil \frac{\mu_{\min}}{2} - 1 \rceil, \lfloor \frac{\mu_{\min}}{2} \rfloor$}. Therefore, there may be better methods available that can match the lower bound and tolerate more Byzantine neighbors than CG$^+$. I suggest the authors discuss whether this gap is fundamental or if CG$^+$ could potentially be improved to match the lower bound exactly.

2. In Corollary 2, the authors only demonstrate the optimality of the proposed CG$^+$ method. I would like to know the theoretical consensus rate of the honest models when using this method. Can the honest models achieve consensus by the end of the training? Could the authors provide theoretical guarantees on how quickly or to what degree consensus is achieved among honest nodes in the training process?

3. The proposed CG$^+$ method does not demonstrate any performance improvement over the existing ClippedGossip in the experiments, which raises doubts about the practical effectiveness of CG$^+$.

4. Equation (CG) appears to be not correct. The update rule of ClippedGossip involves a doubly-stochastic mixing matrix to aggregate messages from neighbors, while (CG) does not include such a mixing matrix. Please refer to (He et al., 2022) and correct this equation.

**Questions:**

My detailed questions are listed in the above section; please refer to it.

---

> ### Author Response · Authors · 2024-11-20
> **Answer to Reviewer**
>
> We thank you for your detailed review and raising some very good questions. We provide a point-by-point response. We would be happy to provide any further clarification if needed.
>
> **On corollary 2**
> In corollary 2 we presented two regimes:
> - the first one where only one step of communication is performed between each optimization step. In this regime honest parameters achieve consensus at a rate of $\mathrm{Va}r_{\mathcal{H}}(x_i^t) \in \mathcal{O}(\frac{1}{T}(1 + \frac{\zeta^2}{\sigma^2})$. This is a direct consequence of the analysis of [2], just as the proof of Corollary 2.
> We have added this result in our paper.
> - In the second regime, close consensus is enforced by multiple communication steps between each optimization step.
>
> **On bi-stochastic matrices**
>
> There are two different approaches for defining gossip matrices, either by using bistochastic matrices (say B), or by considering non-negative symmetric matrices with kernels restricted to the constant vector (say, W). It is always possible to go from one definition to the other and back, for instance using $B = I - W / \lambda$ where $\lambda$ is the largest eigenvalue of $W$. In the paper, we instantiated all algorithms with the Laplacian matrix as the non-negative gossip matrix. Thus, our writing of ClippedGossip and the one of [1] differs only by this definition of gossip matrix and our specific choice of matrix. We felt the "Laplacian" version was more natural since we essentially clip differences along edges.
>
> We understand that the gap between the writing of [1] with a bistochastic matrix and the choice of a non-negative gossip matrix might be confusing, so
> 1) We provided in appendix a note on the links between bi-stochastic and non-negative gossip matrices
> 2) We generalized our paper to Laplacian matrices with arbitrary weights for each edge of the corresponding graph. As such the writing of ClippedGossip from [1] is equivalent to ours.
>
> **On tightness**
>
> First, we must point out that we noted a small error in our proof (see global answer), which we fixed with the following consequences:
> 1) CG+ consists in clipping 2b neighbors per honest node, instead of b+1.
> 2) Our breakdown point is suboptimal by a multiplicative factor of 2, instead of an additive one. Still CG+ outperforms our gossip version of NNA and ClippedGossip (see global answer)
> 3) We performed new experiments, implementing the small changes in CG+. In these ones, CG+ appears to work just as well - or even better - than ClippedGossip.
>
> Considering the sub-optimality by a constant factor, we provided a result stating that it is possible to match exactly the lower bound using CG+ when the clipping threshold can be computed in an oracle way - similarly to what is done in [1] - i.e by defining the clipping threshold by using the honest neighbors parameters only.
>
> A looser assumption giving the same results is that each honest node can identify a subset of 2b neighbors with exactly b Byzantine and b honest. This latter assumption is realistic in the setting of the counterexample in the proof of Theorem 2.
> Hence it is unclear to us whether this gap of a factor 2 is an artifact of CG+ and its analysis or an artifact of our upper bound.
>
> Eventually we point out that if we are suboptimal by a factor 2, previous work [1] on robust gossip algorithms were suboptimal by an unspecified constant factor (equal to 2^10 in the first versions of the paper), divided by the spectral gap of the graph, which can grows as much as the squared number of nodes in the graph (in the case of a line graph). This motivates our claim for near-optimality, since results are often dubbed “optimal” in optimization if they are order-optimal (ignoring constant factors). Yet, we understand that constant factors are important when dealing with robustness, which is why we did our best to obtain such a small gap.
>
> We will add this discussion in our paper.
>
> **On Experiments**
>
> We emphasize that the main goal of this paper is to obtain *theoretical convergence guarantees* for robust gossip algorithms. Notably, the rule used for ClippedGossip in the experiments does not have any theoretical foundation, and the theoretical-supported rule is not implementable, (and the  rate provided for it in the literature is much worse than the new one we get).
>
> However, we have performed during rebuttal extensive experiments on the MNIST dataset and Cifar-10 dataset. We show that CG+ works just as well as ClippedGossip, while being theoretically grounded.
>
> We insist on the fact that having theoretical guarantees is of utter importance when discussing robustness, since experiments are only on a specific dataset against a specific attack. Most real applications require to be sure the methods will not break against new attackers.
>
> [1]: Lie He, Sai Praneeth Karimireddy, and Martin Jaggi. Byzantine-robust decentralized learning via clippedgossip
>
> [2]: Farhadkhani et al. Robust collaborative learning with linear gradient overhead

---

> > ### Comment · Reviewer_jJqZ · 2024-11-26
> >
> > I appreciate the author’s detailed explanation and am satisfied with most of the responses to my comments. However, I believe the newly introduced (RGA) equation does not include the ClippedGossip aggregator as a special case. The weights in (RGA) come from a Laplacian matrix, whereas the ClippedGossip aggregator relies on a doubly stochastic matrix. I do not think these are equivalent. I suggest the authors clarify this point, and if they can address it, I will consider increasing my score.

---

> > > ### Author Response · Authors · 2024-11-27
> > > **Formulation of ClippedGossip as (RGA)**
> > >
> > > Thank you for your answer.
> > >
> > > We precise here how one can write ClippedGossip as (RGA): Let's consider a symmetric bistochastic matrix $B \in [0,1]^{m\times m}$, where $m$ is the total number of nodes in the graph. Based on [1], ClippedGossip writes.
> > >
> > > \begin{align*}
> > > x_i^{t+1} &= \sum_{j=1}^m B_{ij}\left(x_i^t + \mathrm{Clip}(x_j^t - x_i^t, \tau_i)\right)\\
> > > &= x_i^t + \sum_{j=1}^m B_{ij}\mathrm{Clip}(x_j^t - x_i^t, \tau_i)
> > > \end{align*}
> > >
> > > Where we used that each row of $B$ sum to $1$. For any $i\neq j$, we take $B_{ij}$ as the weight of the edge $(i,j)$ in the graph, denoted $w_{ij}$ in (RGA) equation. Note that $B_{ij}=0$ when $i$ and $j$ are not neighbors. Then, considering that the $\mathrm{Rob}$ operator is the $\mathrm{Clip}$ operator, ClippedGossip writes:
> > >
> > > \begin{align*}
> > > x_i^{t+1} = x_i^t + \sum_{j \in n(i)}^m w_{ij}\mathrm{Rob}(x_j^t - x_i^t, \tau_i).
> > > \end{align*}
> > >
> > > This corresponds exactly to our (RGA) equation with communication step size $\eta = 1$. Here the laplacian matrix of the graph writes $L = I - B$. In fact, this shows that the bistochasticity requirement from ClippedGossip can be relieved by removing the $B_{ij}$ in front of the $x_i^t$ factor, and adding proper normalization (in the form of our $\eta$).
> > >
> > > Does this precision satisfy your concern? We would be glad to add any further precisions.
> > >
> > > PS: In the answer to Reviewer mDqu, we establish clear results linking the breakdown points of (RGA) using positive semi-definite symmetric gossip matrices and using bistochastic ones.

---

> > > > ### Comment · Reviewer_jJqZ · 2024-11-27
> > > >
> > > > Thank you for your detailed explanation. As I promise, I increase my score to 6.

---

### Official Review · Reviewer_twSj · 2024-11-04

**Soundness:** 3
**Presentation:** 3
**Contribution:** 3
**Rating:** 6
**Confidence:** 3

**Summary:**

The authors prove a theoretical upper bound on the breakdown point for Byzantine-robust distributed machine learning in decentralized frameworks, and then propose a novel method called $CG^+$, which can achieve the proven upper bound. The theoretical convergence guarantee of $CG^+$ is provided together with empirical evaluation in this paper.

**Strengths:**

+ The problem of obtaining Byzantine robustness in distributed learning on a decentralized framework is challenging and meaningful.
+ This paper is generally well organized.
+ This paper proves a new upper bound of the breakdown point and proposes a novel method, which can reach the optimal breakdown point.

**Weaknesses:**

- The proposed method $CG^+$ is like a combination of ClippedGossip and NNA, and thus the novelty of the proposed method is a little bit limited (I understand that $CG^+$ has a better performance and guarantee than each of the two methods).
- Could the authors briefly explain why the clipping scheme in $CG^+$ can achieve a better theoretical guarantee than ClippedGossip?
- There are replicated references (lines 577-583).

**Questions:**

Please see the weaknesses.

---

> ### Author Response · Authors · 2024-11-20
> **Answer to reviewer**
>
> We thank you for your detailed review and support of the paper. We hereafter provide some answers. Let us know if any further clarification is needed.
>
> 1. A point we did not emphasize enough in our paper is that NNA as stated in [2] is not a gossip algorithm and requires that all nodes communicate together. In this sense, it is not fully decentralized. What we call NNA in our paper is a completely new algorithm in the sense that we adapted NNA to gossip communications, and we derived a complete new analysis for it. As such, the right existing comparisons for CG+ are an essentially centralized algorithm (NNA) and an algorithm which requires oracle knowledge of the identity of nodes (ClippedGossip). In particular, our contribution goes further than “just” mixing the two and obtaining slightly better guarantees.
>
> 2. Another key contribution is the new fully-decentralized attack on gossip algorithms (spectral heterogeneity). This is a state-of-the-art attack which could have been proposed by itself in an independent paper, as it is often the case (see [3, 4]). We provided further experiments in the revision to support the soundness of this attack and exposed that it is widely superior to other standard attacks in the sense that it makes algorithms break sooner.
>
>
> 3. CG+ can achieve better convergence rates than ClippedGossip for two reasons:
>
> - The first one is that their clipping rule relies on an upper bound which is loose when some honest neighbors are significantly farther away from the honest node parameter than the clipping threshold. On the opposite, our upper bound is tight independently of this.
> For instance, if, as [1] do, we compute the clipping threshold using only honest node parameters, we would even gain a multiplicative factor 2 on our breakdown point (which means matching exactly the lower bound).
>
> - The second reason is that the proof of [1] is not tight, and actually even incorrect: the beginning of the proof of Lemma 10 relies on a reversed Jensen inequality.
> Yet this is not a fundamental flaw of the ClippedGossip algorithm itself: using their oracle clipping rule, we obtain with our proof techniques the same convergence results as our gossip version of NNA. Following your concern, we added a theorem stating that ClippedGossip with its oracle clipping rule performs just as good as our gossip version of NNA, i.e lose a multiplicative factor 2 in the breakdown point with respect to CG+. This theorem follows directly from our proof, and fixing the theory of ClippedGossip is therefore another of our contributions.
>
>
> [1]: He, Karimireddy, and Jaggi. Byzantine-robust decentralized learning via clippedgossip.
>
> [2]:  Farhadkhani, Guerraoui, Gupta, Pinot, and Stephan. Byzantine machine learning made easy by resilient averaging of momentums.
>
> [3] Gilad Baruch, Moran Baruch, and Yoav Goldberg. A little is enough: Circumventing defenses for distributed learning. NEURIPS 2019.
>
> [4] Cong Xie, Oluwasanmi Koyejo, and Indranil Gupta. Fall of empires: Breaking byzantine-tolerant sgd by inner product manipulation. In UAI 2020.

---

> > ### Comment · Reviewer_twSj · 2024-11-26
> >
> > I thank the authors for the detailed response, and would like to keep my rating of 6.

---

### Author Response · Authors · 2024-11-20
**Global Answer**

We thank the reviewers for their thoughtful evaluation and time spent evaluating our paper and engaging in the discussions, which greatly helped us frame our work.

We realized that the current writing did not draw a clear line between existing results and our contributions beyond CG+. We have therefore slightly **repositioned the paper as a unifying framework** with new guarantees for several algorithms (new and old ones, in particular but not limited to CG+, a new algorithm with tighter guarantees). We believe this better reflects what this paper brings to the community, without essentially changing the results.

More specifically, we feel that some of our contributions have not been fully recognized, in particular regarding the improvements that we make to existing approaches.
1) **Unified analysis of robust gossip algorithms.** This allows us to derive tight convergence guarantees for both ClippedGossip and a new algorithm based on NNA, which we adapt in the gossip setting. In the light of this unified analysis, we introduce a new algorithm, CG+, which features characteristics from the two others to obtain the strongest robustness guarantees.
2) **Breakdown Point Characterization.** Our upper bound on the breakdown point allows us to characterize the distance between these guarantees and the optimal breakdown point.
3) **Fully decentralized attack**. We test these algorithms by introducing the first attack on gossip algorithms with mathematical foundations. Its effectiveness is proven by experimental results, as it makes aggregation rules break before all the other attacks.


We summarize our results in the following table, which is an extended version of the table present in the original paper (cf. below for changes consecutive to the discussion with reviewers).

|Index | Status |Algorithm      |  Setup   | Breakdown point | Experiments |
| ------ | ------ | ------ | --- | -------- | -------- |
|1| Existing [1] | ClippedGossip w. *oracle rule* | Gossip - not implementable | $b \le \mathcal{O}(\gamma \mu_{\min})$ | none |
|2| Existing [1] | ClippedGossip w. adaptive rule | Gossip | No guarantee | Competitive|
|3| Existing [2] | NNA | Centralized case only |  No guarantee for sparse graphs | None on sparse graphs |
|4| **New (theory)** | Clipped Gossip w. *oracle rule* | Gossip - not implementable | $b \le \mu_{\min}/8$ | none |
|5| **New (algo + theory)** | Gossip NNA | Gossip - practical rule | $b \le \mu_{\min}/8$ | Competitive yet small breakdown |
|6| **New (algo + theory)** | CG+ | Gossip + practical clipping rule | $b\le \mu_{\min}/4$ | Competitive |
|7| **New (algo + theory)** | CG+, *oracle rule* | Gossip - not implementable | $b \le \mu_{\min}/2$ (optimal) | none |

**Summary of the table above and our contributions**

- We obtain the first gossip type algorithms with theoretical guarantees (lines 4 to 6).
- Our CG+ algorithm with practical rule is almost optimal - we rely on a new analysis to obtain this result (line 6).
- If an oracle was permitted for the clipping threshold:
  - We provide a new analysis of ClippedGossip with oracle rule improving on previous literature (line 4 vs. line 1)
  - We provide an optimal rate for our new rule (line 7 vs line 1)
- We define a gossip version of NNA and provide an analysis (line 5 vs line 3)
- In experiments we compare lines 2, 5 and 6. We conclude that both three methods are competitive bellow their own breakdown point. NNA breaks at first, then ClippedGossip. Surprisingly CG+ does not breaks on the MNIST task, even after the theoretical breakdown point, which might be due to the experimental setup, specifically the line search performed to scale the attacks.


[1]: Lie He, Sai Praneeth Karimireddy, and Martin Jaggi. Byzantine-robust decentralized learning via clippedgossip

[2]:  Sadegh Farhadkhani, Rachid Guerraoui, Nirupam Gupta, Le Nguyen Hoang, Rafael Pinot, and John Stephan. Robust collaborative learning with linear gradient overhead. In International Conference on Machine Learning, pp. 9761–9813. PMLR, 2023.

---

> ### Author Response · Authors · 2024-11-20
> **Changes**
>
> **Other Changes(Lemma 7)**
> We identified a small issue within the proof of Lemma 7, and fixed it. The new lemma reads:
>    > The error due to removing honest nodes and due to Byzantine nodes is controlled by the heterogeneity as measured by the gossip matrix.
>
> $\\|E^t\\|^2\le8b\\|X_H^t\\|_{W_H}^2 $
>
> $
> \qquad \quad = 4b\sum_{i \\in H, j \in n_{H}(i)} w_{ij} \\|x_i^t - x_j^t\\|^2 $
>
>
>
> While before the upper bound was 2b.  This does not change much the statements of the theorems, but we actually need to clip $2b$ neighbors instead of $b+1$, which implies that our breakdown guarantee is a factor 2 away from optimum (instead of an additive one). However, we still match the optimum by considering an oracle clipping rule (in the same sense as for [He et al 2022]).
>
>
> **Overall modifications**
> Considering the discussions with reviewers, we made the following changes in the paper:
> - We added extensive experiments on the performance of ClippedGossip, our gossip version of NNA, and CG+ on the MNIST task of the paper, with a varying number of Byzantine neighbors. We conducted experiments on CIFAR-10 as well.
> - Due to the remark of R2 and R3, we generalized the gossip update to Laplacian matrices with generic weights for each edge of the graph. We added a note to enlighten the link between such gossip matrices and the bistochastic matrices as defined in [Koloskova et al]. We had chosen not to do this in the first version to ease reading, especially when stating clipping rules theorem, but we agree that the contribution is more thorough this way.
> - We updated CG+ results and the proofs with the 2b clipping rule, instead of b+1.
> - We added the analysis of ClippedGossip with an oracle clipping threshold.
>
> There is still a little bit of work due on the current revision, and in particular we are slightly over the page limit (by a few lines), but we wanted to provide it as soon as possible to allow some time for discussions. We look forward to interacting with all of you to further clarify certain aspects.

---

### Comment · Area_Chair_vun2 · 2024-11-26
**Response**

Dear Reviewers,

The authors have provided their rebuttal to your questions/comments. It will be very helpful if you can take a look at their responses and provide any further comments/updated review, if you have not already done so.

Thanks!

---

### Meta-Review · Area_Chair_vun2 · 2024-12-20

**Metareview:**

This paper considers a distributed computing set up where devices communicate with each other directly, however are subject to a fraction of them being Byzantine. They show an upper bound on the number of tolerable adversaries for a gossip-algorithm to still be able to find a solution.

While the paper is interesting, two of the reviewers are on the fence, while one suggested rejection. The main complaint seem to be that of a lack of clarity and comparison with available results. Referees also raise the issue of lack of experimental validations. Reviewer mDqu
engaged with the authors and after a log discussion decided to keep their score.

Based on the comments and the discussion, I recommend rejection of the article at this stage.

**Additional Comments On Reviewer Discussion:**

Reviewers took part in discussion

---

### Decision · Program_Chairs · 2025-01-22

Reject